# Contrastive Consolidation of Top-Down Modulations Achieves Sparsely Supervised Continual Learning

**Viet Anh Khoa Tran**
Peter Grünberg Institute
Forschungszentrum Jülich & RWTH Aachen
v.tran@fz-juelich.de

**Emre Neftci**
Peter Grünberg Institute
Forschungszentrum Jülich & RWTH Aachen
e.neftci@fz-juelich.de

**Willem A. M. Wybo**
Peter Grünberg Institute
Forschungszentrum Jülich
w.wybo@fz-juelich.de

## Abstract

Biological brains learn continually from a stream of unlabeled data, while integrating specialized information from sparsely labeled examples without compromising their ability to generalize. Meanwhile, machine learning methods are susceptible to catastrophic forgetting in this natural learning setting, as supervised specialist fine-tuning degrades performance on the original task. We introduce **task-modulated contrastive learning (TMCL)**, which takes inspiration from the biophysical machinery in the neocortex, using predictive coding principles to integrate top-down information continually and without supervision. We follow the idea that these principles build a view-invariant representation space, and that this can be implemented using a contrastive loss. Then, whenever labeled samples of a new class occur, new affine modulations are learned that improve separation of the new class from all others, without affecting feedforward weights. By co-opting the view-invariance learning mechanism, we then train feedforward weights to match the unmodulated representation of a data sample to its modulated counterparts. This introduces modulation invariance into the representation space, and, by also using past modulations, stabilizes it. Our experiments show improvements in both class-incremental and transfer learning over state-of-the-art unsupervised approaches, as well as over comparable supervised approaches, using as few as 1% of available labels. Taken together, our work suggests that top-down modulations play a crucial role in balancing stability and plasticity.

## 1 Introduction

Input data streams encountered by animals or humans during development differ markedly from those commonly used in machine learning. In contemporary machine learning (e.g. foundation models), data streams typically consist of unlabeled data, augmented with some degree of supervised fine-tuning in the final training stages [1]. Such an approach is difficult to translate to the continual learning setting encountered in natural data streams, as the naive introduction of fine-tuning stages often leads to catastrophic forgetting [2–4]. Meanwhile, animals and humans receive mostly unsupervised inputs, interspersed with sparse supervised data, which could, for instance, be provided through an external teacher (e.g. a parent telling their child that the object is called an 'apple'). Compared to unsupervised data streams, such sparse supervisory episodes are infrequent. Therefore, the learning dilemma that

39th Conference on Neural Information Processing Systems (NeurIPS 2025).

arises is how continual learning algorithms can benefit from sparse supervisory episodes without negatively affecting representations learned in an unsupervised manner.

Here, we draw inspiration from the circuitry in biological brains to solve this learning dilemma. We leverage the fact that cortical neurons can – broadly speaking – be subdivided into a proximal, perisomatic zone, receiving feedforward inputs [5–7], and a distal, apical region, receiving top-down modulatory inputs [6–11] (Figure 1, left). Through their physical separation, these zones are functionally distinct [12], and implement different plasticity principles [13, 14]. Learning of the perisomatic, feedforward connections is believed to follow a form of predictive coding [15, 16] that is the biological analogue of self-supervised learning such as VICReg [17] or CPC [18]. At the same time, top-down modulations to distal dendrites provide a contextual, modulatory signal to the feedforward network [19–26]. We hypothesize that this signal is learned during the supervised learning episodes. In machine learning, this concept has been explored in the context of parameter-efficient fine-tuning by training task-specific scaling and/or shifting terms [27–32]. This provides a straightforward solution to continual learning problems for which the task identity is known during both training and evaluation (i.e. task-incremental learning [33]), as modulations for new tasks can be learned without affecting core feedforward weights [34–36]. However, when the task identity is not known at evaluation (i.e. class-incremental learning), the modulated representations for each task need to be consolidated in a shared representation space.

To achieve class-incremental learning, we hypothesize, based on the spatial and functional segregation of distal dendrites, that the top-down signal is not affected by the predictive plasticity of feedforward weights in the perisomatic region. As such, it leaves a permanent imprint on the network that, through occasional reactivation, integrates the new percept in the neural representation space, while also providing a form of functional regularization that limits forgetting. We demonstrate that these effects are achieved by standard predictive coding principles that proceed over modulated representations. In our task-modulated contrastive learning (TMCL) algorithm, we use the currently available labeled examples for each new class to learn modulations that *orthogonalize* their representations from all others. These modulations are then frozen and applied to the network during the feedforward weight learning, using only currently available unlabeled samples, and effectively *consolidate* the task-specific knowledge encoded in the modulations into the feedforward weights (Figure 1, middle). This departs from the conventional pretraining-finetuning paradigm, where naive reintegration of specialized models into the general one causes catastrophic forgetting [1, 4] (Figure 1, right).

We evaluate the performance of TMCL on the standard class-incremental CIFAR-100 benchmark, outperforming state-of-the-art purely unsupervised, purely supervised, and hybrid approaches in label-scarce scenarios. Furthermore, we evaluate its transfer learning capabilities across a diverse set of downstream tasks, demonstrating its effectiveness in learning generalizable representations that extend beyond adaptation to CIFAR-100. Finally, we show that our method dynamically navigates the stability-plasticity dilemma through adaptation of the consolidation term.

## 2  Related Work

**Biological representation learning.**   Several authors have explored the idea that the cortex learns in a self-supervised manner [15, 16, 37–40]. Although complementary approaches based on adversarial samples have also been proposed [37], most theories focus on some form of predictive coding, where the cortex learns to predict the next inputs given the current neural representation. Mikulasch et al. [38, 39] take a classic view on this, where a loss function to the next layer reconstructs the input, while Kermani Nejad et al. [40] theorize that the architecture of the cortical microcircuit is well-suited for predictive coding. Finally, Illing et al. [15] and Halvagal and Zenke [16] propose local plasticity rules based on, respectively, CPC [18] and VICReg [17] that, as they argue, in a natural setting could proceed by comparing neural representations at subsequent time steps. We extend this idea with an explanation of how top-down modulations could be incorporated into the learning process.

**Learning with modulations.**   The expressivity of learning modulations was initially demonstrated by Perez et al. [27] to solve visual reasoning problems. Frankle et al. [28] subsequently showed that a surprisingly high performance can be achieved with ResNets [41] while just training BatchNorm layers, which — if performed per-task — is equivalent to learning affine modulations. Finally, it was shown simultaneously in language and vision that fine-tuning through modulations in transformer

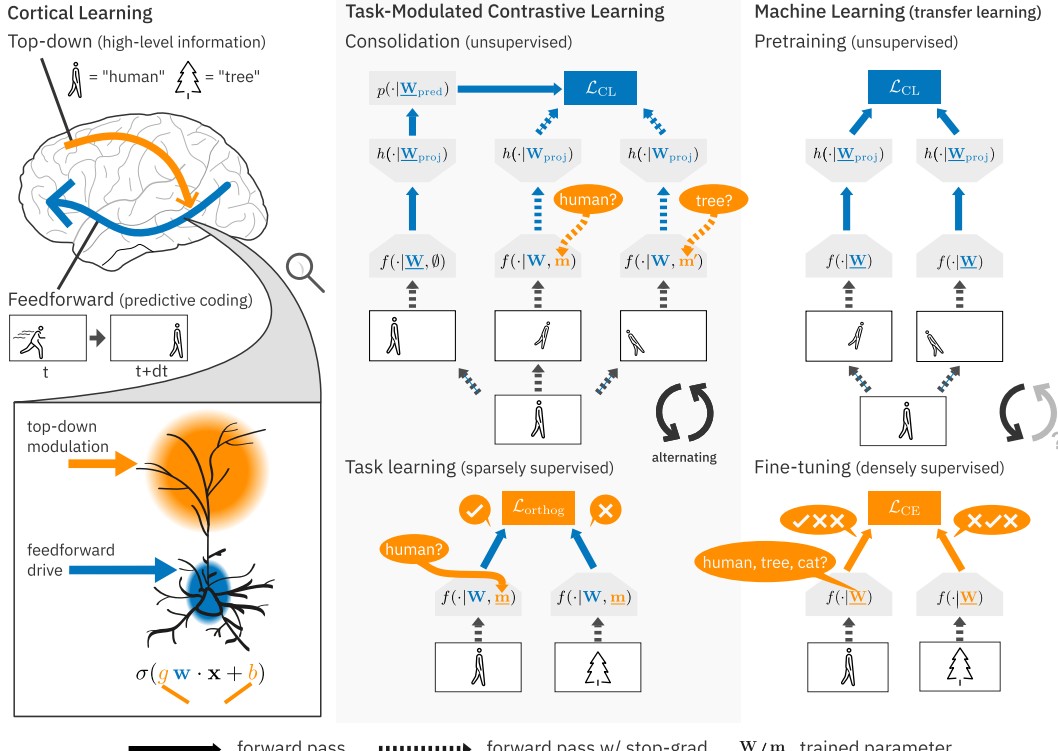

Figure 1: **Biologically inspired consolidation of high-level modulations into feedforward weights.** Cortical learning (**left**) is characterized by the interplay between top-down (orange) and feedforward (blue) processing, where top-down connections impart high-level information on the feedforward sensory processing pathway (top). The feedforward pathway, on the other hand, learns to predict neural representations of future inputs (predictive coding). Notably, top-down and feedforward information arrives at spatially segregated loci on sensory neurons (bottom), suggesting distinct roles in shaping the neuronal input-output relation (cf. [19]) as well as distinct plasticity processes governing weight changes. Translating this view to a machine learning algorithm (**middle**), we *(i)* train modulations to implement high-level object identification tasks as the analogue of top-down inputs (bottom, solid arrows, but not dashed ones, indicate that gradients backpropagate in the opposite direction, and underlined parameters are trained), while we *(ii)* train for view invariance over modulated representations – and thus also for modulation invariance – as the analogue of predictive coding (top). As a consequence, high-level information continually permeates into the sensory processing pathway, which can be contrasted with the traditional machine learning (**right**) approach of unsupervised pretraining for view invariance (top) followed by supervised fine-tuning (bottom). In this case, it is unclear how high-level information can be incorporated into the sensory processing pathway to improve subsequent learning.

models is particularly powerful, as it reaches the same performance as using all parameters [29, 30, 32].

Modulations are an attractive way to implement task-incremental learning, as task-specific modulations can be learned for each new task without affecting feedforward weights. Masse et al. [34] propose gating random subsets of neurons, whereas Iyer et al. [36] provide a biological interpretation. Fine-tuning through modulations [30, 32] can also be considered as a form of continual learning, as it can be applied in sequence to any new dataset.

**Continual representation learning.** Traditionally, continual learning has focused on purely supervised methods [42–67]. These methods can be categorized into replay-based approaches [42–50], regularization-based approaches [50–62, 67] and approaches introducing new parameters [63–66]. TMCL can be considered a regularization-based approach, but it also introduces new parameters. However, these parameters are not used during inference. Recently, purely self-supervised con-

tinual learning algorithms have been proposed [68–72], where state-of-the-art algorithms [69, 72] predict past representations from a stored model copy without exemplar replay. Very recently, semi-supervised continual learning approaches have emerged [73–76], which consolidate by distilling from expert models [76, 77] or for which the labels are only used for readout learning [74].

We highlight SIESTA [49], CLS-ER [78] and DualNet [73], which are inspired by the complementary learning systems theory (CLS) [79, 80], the idea that learning occurs at *fast* (task-learning) and *slow* (consolidation) timeframes. However, all these methods interpret CLS to assume sample replay provided as episodic memory via the hippocampus. Instead, we suggest functional replay of task modulations ('How did we solve the task?'), and do not investigate methods with exemplar replay. Still, we point out similarities to the replay-based DualNet, which introduces a fast supervised network generating modulations on top of a slow self-supervised network. However, DualNet consolidates only as new labels arrive. Consolidation in TMCL requires no labels, instead exploiting previously learned task modulations.

## 3   Modulation-Invariant Continual Representation Learning

We follow the idea from Iyer et al. [36] that cortical networks learn to interpret novel information by learning new top-down modulations, and propose that consolidation of these modulations is a crucial component of learning in biological brains. This motivates our task-modulated contrastive learning (TMCL) algorithm as the machine learning analogue of this consolidation, tackling continual representation learning. We consider the parameters of conventional machine learning models as *task-agnostic* feedforward weights $\mathbf{W}$. On top of these weights, we introduce per-task affine transformation parameters as *task-specific* modulations $\mathbf{m}_t$, the analogue of biological top-down modulations. We denote the modulated network as $f(\mathbf{x}|\mathbf{W}, \mathbf{m})$ with feedforward weights $\mathbf{W}$ and modulations $\mathbf{m}$, while $f(\mathbf{x}|\mathbf{W}, \emptyset)$ represents the unmodulated network (i.e. where the modulations are identity operations).

In TMCL, the overall objective is to arrive at an unmodulated representation space where all classes $c \in \mathcal{C}$ in the dataset $\mathcal{D}$ have compact representations clustered around mutually orthogonal class centers, i.e.

$$\gamma^c \perp \gamma^{c'}, \forall c, c' \in \mathcal{C}, \tag{1}$$

with $\gamma^c = \mathbb{E}_{\mathbf{x} \in \mathbf{X}^{(c)}}[f(\mathbf{x}|W, \emptyset)]$, where $\mathbf{X}^{(c)}$ is the set of samples from class $c$. Because we assume a continual learning setting, where we do not have all class samples at our disposal, we do not optimize for (1) directly. Rather, we achieve this by breaking the optimisation procedure down into two distinct learning objectives. The first objective (Figure 2, bottom left) is to orthogonalize any given class $c$ from the others in a *modulated* representation space, i.e. we learn a modulation $\mathbf{m}^c$ so that the class center of class $c$ becomes orthogonal to all other classes in the modulated space:

$$\gamma^c_{\mathbf{m}^c} \perp \{\gamma^{c'}_{\mathbf{m}^c} : c' \in \mathcal{C} \setminus \{c\}\}, \tag{2}$$

where $\gamma^c_{\mathbf{m}}$ denotes the representation of the class center under modulation $\mathbf{m}$, i.e. $\gamma^c_{\mathbf{m}} = \mathbb{E}_{\mathbf{x} \in \mathbf{X}^{(c)}}(f(\mathbf{x}|W, \mathbf{m}))$. The second objective (Figure 2, bottom right) is entirely unsupervised and trains network weights to become *modulation-invariant*, so that

$$\gamma^c = \gamma^c_{\mathbf{m}^{c'}} \forall c' \in \mathcal{C}. \tag{3}$$

It can be seen that a representation space that satisfies *both* (2) and (3), also satisfies (1).

In our continual setting, which we adapt from Fini et al. [69], training is partitioned into $s \in 1, \ldots, S$ sessions, so that (1) can only be achieved approximately. In each session $s$, we only observe unlabeled samples $x \in \mathcal{D}^{(s)} \subset \mathcal{D}$ belonging to the session-specific partition of classes $\mathcal{C}^{(s)} \subset \mathcal{C}$. Additionally, a fraction of labeled samples $(x, y) \in \mathcal{D}^{(s)}_{\text{sup}} \subset \mathcal{D}^{(s)}$ is made available to (3). As a consequence, during each session (Figure 2), we learn objective (2) restricted to $\mathcal{D}^{(s)}_{\text{sup}}$ in a first phase, and then learn objective (3) using unlabeled samples from $\mathcal{D}^{(s)}$. We explain the implementation of both phases in detail below.

**Learning modulations that orthogonalize new class representations.**   Whenever a new class label is observed, we learn class modulations on top of frozen feedforward weights to implement objective (2), which improves separation of the new class representations from all others currently

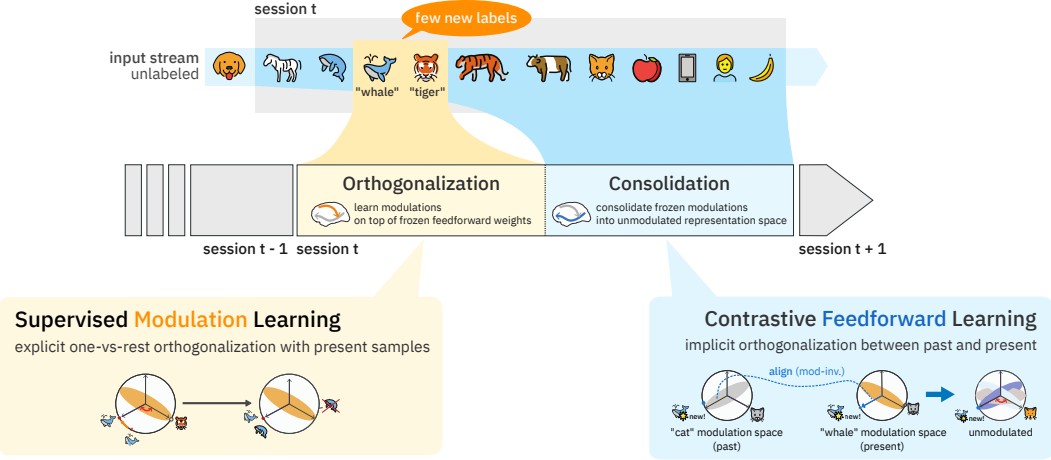

Figure 2: **Sparsely labeled class-incremental representation learning.** We implement continual learning over mostly unlabeled data streams, where only a few labeled samples are provided (**top**). To give an intuition of our algorithm (**bottom**), we consider that after successfully incorporating the data seen thus far, sufficiently collapsed neural representations exist for the already seen data classes after session $t-1$ (here dog, cat). For a new data class in session $t$ (e.g. whale), such a collapsed representation may not yet exist. We then learn a *new* set of modulations to collapse "whale" representations in the modulated representation space, *orthogonalizing* them from all other available labeled examples, thus obtaining an orthogonal subspace for everything that is non-whale. Then, occasional reactivation of the "whale" modulation in $\mathcal{L}_{\mathrm{CL}}$ draws unmodulated "whale" representations towards this collapsed representation (cf. Figure 1, middle), while drawing other samples to the orthogonal subspace, thus *consolidating* "whale" into the unmodulated representation space.

available. We assume that explicit class labels for these other classes are not available, therefore, the conventional machine learning approach of all-vs-all classification is not applicable. Instead, we learn one-vs-rest class modulations, only using currently available samples as negatives (Figure 2, bottom left). Note that if negative samples were to collapse to a single representation, (2) and (3) could not hold simultaneously and therefore (1) could not be achieved either. For this reason, we apply a variation of the **orthogonal projection loss** (OPL) [81] instead of binary cross-entropy (BCE) [82]. We define $s_{\mathbf{m}}(\mathbf{u},\mathbf{v}) = \mathrm{sim}(f(\mathbf{u}|\mathbf{W},\mathbf{m}), f(\mathbf{v}|\mathbf{W},\mathbf{m}))$ as the cosine similarity between samples $\mathbf{u}$ and $\mathbf{v}$ under modulation $\mathbf{m}$, i.e. $\mathrm{sim}(\mathbf{u},\mathbf{v}) = \frac{\mathbf{u}^T\mathbf{v}}{\|u\|\|v\|}$. Then, given a batch $\mathbf{X}^{(c)}$ of $c$-class examples and a batch $\mathbf{X}^{(\neg c)}$ of non-$c$ examples, we define

$$\mathcal{L}_{\mathrm{OPL}}^{(c)} := \sum_{\mathbf{p},\mathbf{p}'\in\mathbf{X}^{(c)}} \underbrace{(1 - s_{\mathbf{m}}(\mathbf{p},\mathbf{p}'))}_{\text{collapse}} + \sum_{\substack{\mathbf{p}\in\mathbf{X}^{(c)}\\\mathbf{n}\in\mathbf{X}^{(\neg c)}}} \underbrace{|s_{\mathbf{m}}(\mathbf{p},\mathbf{n})|}_{\text{orthogonalization}}. \tag{4}$$

$\mathbf{m}^{(\mathbf{c})}$ is then found as $\min_{\mathbf{m}} \mathcal{L}_{\mathrm{OPL}}^{(c)}$. The second term draws the cosine similarities between class $c$ and non-class $c$ representations to zero, leading to an orthogonalization of class $c$ representations from all others, therefore approximating objective (2).

**Consolidation of modulations into a view- and modulation-invariant representation space.** To implement objective (3), we co-opt self-supervised contrastive learning, which is considered a biological analogue of predictive learning principles of the feedforward connections [15, 16]. Contrastive learning objectives train for view-invariance (VI), as they attract representations of views of the same source sample under different view augmentations to each other, while repelling representations of other samples [17, 18, 83–92]. These view augmentations $\alpha$, forming representations colloquially referred to as 'positives', are sampled randomly from a set of augmentations $\mathcal{A}$ (i.e. $\alpha \sim \mathcal{A}$), which includes combinations of e.g. random crops, color jitter and horizontal flips. We note that modifying the set of positives, while using the same contrastive learning objective, results in different invariances being learned. We generalize the contrastive loss $\mathcal{L}_{\mathrm{CL}}(\{\mathbf{z}_1,\ldots,\mathbf{z}_K\})$ as a generic learning rule

operating on a set of $K$ positives $\mathbf{z}_1, \ldots, \mathbf{z}_K$. Then, we formalize the view-invariant loss as

$$\text{VI} := \mathcal{L}_{\text{CL}}(\{\varphi_{\text{VI}}(\alpha_k(\mathbf{x})|\mathbf{W}, \emptyset) \mid k = 1, \ldots, K, \ \alpha_k \sim \mathcal{A}\}), \tag{5}$$

where $\varphi_{\text{VI}} = h_{\text{VI}} \circ f$ and $h_{\text{VI}}(\cdot)$ is an MLP used exclusively for the view-invariant objective.

To consolidate the orthogonalizing modulations into the unmodulated representation space, we propose to employ *differently modulated* views of the source sample as positives, hence constructing a modulation-invariant representation space (Figure 1, top center). Concretely, we uniformly sample $\mathbf{m}_2, \ldots, \mathbf{m}_V \sim \mathcal{M}^{(s)}$ from the set of trained modulations at session $s$. We formalize the modulation-invariant loss, which implements objective (3), as

$$\begin{aligned}
\text{MI} := \mathcal{L}_{\text{CL}}\Big( & \{p_{\text{MI}}(\varphi_{\text{MI}}(\alpha_1(\mathbf{x})|\mathbf{W}, \emptyset))\} \\
& \cup \Big\{\text{sg}(\varphi_{\text{MI}}(\alpha_k(\mathbf{x})|\mathbf{W}, \mathbf{m}_k)) \ \Big| \ k = 2, \ldots, K, \ \alpha_k \sim \mathcal{A}, \ \mathbf{m}_k \sim \mathcal{M}^{(s)}\Big\}\Big),
\end{aligned} \tag{6}$$

where $\text{sg}(\cdot)$ denotes a stop-gradient, $\varphi_{\text{MI}} = h_{\text{MI}} \circ f$, and $p_{\text{MI}}(\cdot), h_{\text{MI}}(\cdot)$ are MLPs used exclusively for the modulation-invariant objective. Therefore, a compact representation of each new class is *consolidated* in the unmodulated representation space. During the consolidation phase of TMCL, the combination of VI and MI is optimized jointly.

**Mapping prior work to the canonical contrastive loss and comparing to TMCL.** Notably, in supervised contrastive learning (SupCon), samples of the same class are used as positives [93], thus implementing invariance to features not predictive of class identity. SupCon is a straightforward method to additionally leverage the few available labels in semi-supervised continual learning setups. However, in contrast to our MI objective, SupCon only separates classes for which labels are available in the current session, while MI implicitly separates current samples from past class centers.

State-of-the-art unsupervised continual representation learning algorithms such as CaSSLe [69] and PNR [72] train the network representations to be invariant to the model state (state invariance or SI), by using as positives one representation obtained from the current network state and one representation obtained from a stored, past network state. In TMCL, because the modulations for any given class are *not* updated after the initial learning, they effectively constitute an imprint of neural activities that separates that class from all others. Therefore, training the unmodulated representations to maintain similarity with this imprint also stabilizes continual learning, and can be understood as a form of SI that does not require storing the full network state twice (cf. Table A1, right).

## 4 Experiments

**Experimental protocol.** We adopt a standard class-incremental continual learning protocol on both CIFAR-100 and ImageNet-100, dividing the dataset into five sessions, each containing 20 disjoint classes. We additionally introduce a supervised cross-entropy (CE) baseline with a projection head [94], which has been reported to outperform self-supervised methods. For each session, we train the model for 100 orthogonalization epochs, where we train modulations for the new classes, and 200 consolidation epochs. We further start the first session with a pretraining phase of 250 epochs, where the feedforward weights are updated via VI, or analogously with SupCon and CE for the respective supervised methods. For all loss terms except CaSSLe and PNR, we use $K = 4$ positives. The backbone $f$ is a modified ConViT architecture [95] as introduced in DyTox [65]. We emphasize that this architecture is equivalent to ResNet-18 in both memory and compute (Table A1, b). We evaluate the representations via linear probing of the last four layers of $f$ as suggested by Caron et al. [88]. Further details are provided in supplementary materials A.

**Semi-supervised continual representation learning.** In Table 1, we present the all-vs-all linear readout accuracies on the CIFAR-100 and ImageNet-100 datasets after class-incremental learning. As previously reported [94], continual supervised learning outperforms self-supervised learning if all labels are observable, while purely supervised methods significantly degrade with only 10% of labels or fewer. Clearly, a combination of both supervision and self-supervision proves most performant in this setup, as VI + SupCon significantly outperforms the state-of-the-art unsupervised algorithm (VI + SI (PNR)). While *state invariance* is helpful, most of the improvement stems from the additional supervision (VI + SupCon). In the fully labeled scenario, VI + MI improves

Table 1: **Semi-supervised continual representation learning.** Final linear readout accuracy (all-vs-all) on class-incremental ImageNet-100 and CIFAR-100 with 5 sessions (averaged over three and four seeds respectively, $\pm$ denotes the standard deviation).

| Method | CIFAR-100/5 | | | ImageNet-100/5 | | |
|---|---|---|---|---|---|---|
| | 100% | 10% | 1% | 100% | 10% | 1% |
| *Supervised methods* | | | | | | |
| SupCon | $58.4_{\pm0.7}$ | $47.7_{\pm0.9}$ | $39.1_{\pm1.4}$ | $64.0_{\pm0.7}$ | $48.7_{\pm1.2}$ | $33.6_{\pm0.5}$ |
| CE | $60.1_{\pm0.5}$ | $50.7_{\pm0.3}$ | $41.6_{\pm0.5}$ | $66.0_{\pm0.4}$ | $50.1_{\pm1.2}$ | $35.2_{\pm0.2}$ |
| *Self-supervised methods* | | | | | | |
| VI | | $59.3_{\pm0.2}$ | | | $59.7_{\pm0.3}$ | |
| *...integrating class labels (semi-supervised)* | | | | | | |
| + CE | $60.6_{\pm0.6}$ | $59.5_{\pm0.3}$ | $59.8_{\pm0.2}$ | $64.5_{\pm0.7}$ | $61.4_{\pm0.9}$ | $60.3_{\pm0.5}$ |
| + SupCon | $62.2_{\pm0.2}$ | $59.6_{\pm0.4}$ | $59.3_{\pm0.2}$ | $\mathbf{66.6}_{\pm0.4}$ | $61.4_{\pm0.6}$ | $60.2_{\pm0.3}$ |
| + MI (**TMCL**) | $60.7_{\pm0.4}$ | $\mathbf{61.1}_{\pm0.3}$ | $60.7_{\pm0.3}$ | $64.5_{\pm0.3}$ | $\mathbf{63.5}_{\pm0.4}$ | $\mathbf{62.0}_{\pm0.2}$ |
| *...introducing state invariance (and class labels)* | | | | | | |
| + SI (PNR) | | $60.2_{\pm0.2}$ | | | $59.6_{\pm1.0}$ | |
| + CE | $61.2_{\pm0.3}$ | $60.3_{\pm0.6}$ | $60.1_{\pm0.3}$ | $64.5_{\pm0.5}$ | $60.3_{\pm0.8}$ | $59.4_{\pm0.9}$ |
| + SupCon | $\mathbf{62.7}_{\pm0.2}$ | $60.7_{\pm0.3}$ | $60.1_{\pm0.3}$ | $67.0_{\pm0.2}$ | $60.2_{\pm0.3}$ | $58.5_{\pm0.4}$ |
| + MI | $60.9_{\pm0.2}$ | $60.7_{\pm0.2}$ | $\mathbf{60.9}_{\pm0.2}$ | $63.8_{\pm0.9}$ | $62.7_{\pm0.6}$ | $61.7_{\pm0.4}$ |

Table 2: **Transfer learning.** Final all-vs-all kNN accuracy on diverse downstream tasks after five incremental CIFAR-100 sessions (averaged over four seeds, $\pm$ denotes the standard deviation).

| | Method | Aircraft | CIFAR-10 | CUBirds | DTD | EuroSAT | GTSRB | STL-10 | SVHN | VGGFlower |
|---|---|---|---|---|---|---|---|---|---|---|
| *1% CIFAR labels* | SupCon | $8.3_{\pm1.1}$ | $52.0_{\pm2.0}$ | $3.3_{\pm0.3}$ | $15.5_{\pm0.6}$ | $64.1_{\pm3.3}$ | $38.5_{\pm3.5}$ | $44.9_{\pm1.4}$ | $46.6_{\pm2.1}$ | $18.9_{\pm2.0}$ |
| | + SI (CaSSLe) | $11.6_{\pm3.1}$ | $51.9_{\pm1.3}$ | $3.4_{\pm0.3}$ | $16.3_{\pm1.4}$ | $66.4_{\pm4.3}$ | $38.3_{\pm4.9}$ | $44.8_{\pm1.8}$ | $46.3_{\pm1.1}$ | $21.9_{\pm5.3}$ |
| | VI | $27.5_{\pm0.7}$ | $77.0_{\pm0.3}$ | $10.0_{\pm0.2}$ | $27.6_{\pm0.8}$ | $86.0_{\pm0.4}$ | $67.8_{\pm0.2}$ | $65.4_{\pm0.5}$ | $48.3_{\pm0.4}$ | $58.5_{\pm0.6}$ |
| | + SupCon | $27.4_{\pm0.6}$ | $77.3_{\pm0.3}$ | $9.8_{\pm0.2}$ | $27.9_{\pm0.5}$ | $85.8_{\pm0.1}$ | $68.2_{\pm1.3}$ | $64.9_{\pm0.6}$ | $49.8_{\pm0.4}$ | $58.4_{\pm0.6}$ |
| | + MI (**TMCL**) | $28.0_{\pm0.5}$ | $78.0_{\pm0.3}$ | $10.7_{\pm0.2}$ | $29.4_{\pm0.8}$ | $87.1_{\pm0.2}$ | $68.2_{\pm0.9}$ | $66.3_{\pm0.3}$ | $49.0_{\pm0.7}$ | $61.7_{\pm0.5}$ |
| | + SI (PNR) | $28.5_{\pm0.5}$ | $78.3_{\pm0.1}$ | $11.1_{\pm0.1}$ | $28.6_{\pm0.7}$ | $87.0_{\pm0.2}$ | $69.4_{\pm0.4}$ | $67.0_{\pm0.4}$ | $49.1_{\pm0.8}$ | $64.7_{\pm0.5}$ |
| | + SupCon | $29.1_{\pm0.2}$ | $78.3_{\pm0.2}$ | $10.5_{\pm0.4}$ | $28.2_{\pm0.4}$ | $87.0_{\pm0.5}$ | $\mathbf{70.2}_{\pm0.2}$ | $66.8_{\pm0.3}$ | $\mathbf{49.9}_{\pm0.6}$ | $64.4_{\pm0.3}$ |
| | + MI | $\mathbf{29.9}_{\pm0.8}$ | $\mathbf{79.0}_{\pm0.2}$ | $\mathbf{11.8}_{\pm0.3}$ | $29.5_{\pm0.3}$ | $\mathbf{87.5}_{\pm0.2}$ | $69.8_{\pm0.9}$ | $\mathbf{67.6}_{\pm0.3}$ | $49.7_{\pm0.7}$ | $\mathbf{66.2}_{\pm0.4}$ |
| *100% C.l.* | CE [94] | $\mathbf{29.0}_{\pm0.6}$ | $78.4_{\pm0.5}$ | $10.0_{\pm0.1}$ | $28.5_{\pm0.8}$ | $83.4_{\pm0.4}$ | $64.0_{\pm0.4}$ | $66.2_{\pm0.5}$ | $\mathbf{53.2}_{\pm0.6}$ | $58.2_{\pm0.8}$ |
| | VI | $27.5_{\pm0.7}$ | $77.0_{\pm0.3}$ | $10.0_{\pm0.2}$ | $27.6_{\pm0.8}$ | $86.0_{\pm0.4}$ | $67.8_{\pm0.2}$ | $65.4_{\pm0.5}$ | $48.3_{\pm0.4}$ | $58.5_{\pm0.6}$ |
| | + SupCon | $28.1_{\pm0.7}$ | $\mathbf{79.1}_{\pm0.3}$ | $10.4_{\pm0.5}$ | $28.9_{\pm0.7}$ | $86.6_{\pm0.1}$ | $68.6_{\pm0.6}$ | $\mathbf{66.8}_{\pm0.2}$ | $49.8_{\pm0.8}$ | $58.2_{\pm0.4}$ |
| | + MI | $28.9_{\pm0.3}$ | $78.2_{\pm0.2}$ | $\mathbf{10.7}_{\pm0.2}$ | $\mathbf{29.2}_{\pm0.2}$ | $\mathbf{87.0}_{\pm0.1}$ | $\mathbf{71.0}_{\pm0.8}$ | $66.7_{\pm0.3}$ | $50.7_{\pm0.6}$ | $\mathbf{62.8}_{\pm0.5}$ |

over VI + SI, yet underperforms VI + SupCon. This changes as we move towards label-sparse scenarios, where MI consistently outperforms SupCon. Interestingly, MI is not orthogonal to SI, as their combination results in the strongest performances in label-sparse scenarios.

**Representational quality for transfer learning.** In Table 2, we report k-nearest neighbors (kNN) performance of different tasks on top of the CIFAR-100 continually pretrained models, without any further fine-tuning or training. First, on those models that only observe 1% of the training labels (or no labels at all, i.e. VI, VI + SI), we demonstrate that modulation invariance improves representational quality beyond solely adapting to CIFAR-100 classes, as performances across almost all probed datasets improve. Here, MI is not orthogonal to SI, and the combination of VI+SI+MI results in the most transferable representations. Furthermore, amongst methods exploiting CIFAR-100 labels, MI outperforms SupCon, supporting the hypothesis that direct supervision results in representations that are 'greedily' invariant to features which are not relevant for the current task. Strikingly, this cannot be solely attributed to the sparse supervision setup; conducting the same study on models trained with 100% labels, we observe that MI outperforms SupCon, except for CIFAR-10 and STL-10, which are semantically similar to CIFAR-100. Furthermore, semi-supervised VI + MI outperforms fully supervised CE on most tasks, further underlining the lack of generalization of class-based invariance

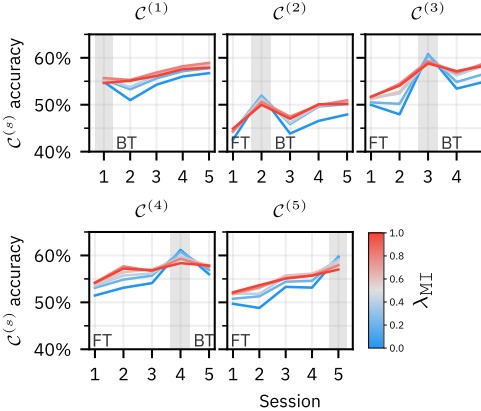

Figure 3: **Controlling the stability-plasticity trade-off.** Accuracy on CIFAR-100 test samples associated with $\mathcal{C}^{(s)}$, measuring forward and backward transfer (averaged over four seeds). Color scale indicates the strength of MI.

Table 4: **Backwards and forwards transfer performances,** using 1% of labels for SupCon and MI (averaged over four seeds, $\pm$ denotes the standard deviation).

| Methods | BT | FT |
|---|---|---|
| SupCon | $-18.5_{\pm 0.4}$ | $-27.7_{\pm 0.4}$ |
| + SI (CaSSLe) | $-18.1_{\pm 0.2}$ | $-27.4_{\pm 0.4}$ |
| VI | $-1.7_{\pm 0.4}$ | $-6.5_{\pm 0.4}$ |
| + MI | $-0.3_{\pm 0.2}$ | $-3.7_{\pm 0.4}$ |
| + SupCon | $-1.9_{\pm 0.1}$ | $-5.9_{\pm 0.3}$ |
| + SI (CaSSLe) | $1.0_{\pm 0.1}$ | $-4.4_{\pm 0.4}$ |
| + MI | $\mathbf{1.3}_{\pm 0.2}$ | $\mathbf{-3.0}_{\pm 0.3}$ |
| + SupCon | $1.0_{\pm 0.1}$ | $-3.7_{\pm 0.1}$ |

learning. In contrast to the CIFAR-100 results (Table 1), we observe that as more labels are provided to MI, representational transfer quality improves for most tasks.

**Robustness to noisy labels.** Erroneous labels are commonly encountered in natural learning environments. While in that case, misclassification of samples would be expected, the quality of the representations should not deteriorate. However, introducing label noise severely degrades performance in fully supervised methods (Table 3, red cells denoting performance inferior to the self-supervised baseline), and naive integration via CE also significantly underperforms the self-supervised baseline. While VI + SupCon only degrades with as much as 90% label noise, TMCL is robust to label noise up to 99% and does not underperform the self-supervised baseline. We hypothesize that this robustness arises because in TMCL, the erroneous labels do not directly affect weight learning. Rather, they lead to nonsensical modulations, which represent a noise contribution to which the contrastive learning machinery learns to become invariant.

Table 3: **Label noise.** Final linear readout accuracy on continual CIFAR-100, replacing a fraction of labels with random labels. (averaged over four seeds, $\pm$ denotes the standard deviation).

| Method | Label noise | | | |
|---|---|---|---|---|
| | **30%** | **50%** | **90%** | **99%** |
| *Fully supervised methods* | | | | |
| CE | $57.3_{\pm 0.2}$ | $54.4_{\pm 0.2}$ | $8.2_{\pm 1.2}$ | $6.4_{\pm 2.2}$ |
| SupCon | $58.0_{\pm 0.4}$ | $56.3_{\pm 0.3}$ | $48.6_{\pm 1.0}$ | $47.0_{\pm 0.6}$ |
| *Self-supervised baseline* | | | | |
| VI | $59.3_{\pm 0.2}$ | | | |
| *...integrating (noisy) labels* | | | | |
| + CE | $59.2_{\pm 0.6}$ | $59.2_{\pm 0.2}$ | $58.3_{\pm 0.4}$ | $58.1_{\pm 0.5}$ |
| + SupCon | $\mathbf{60.6}_{\pm 0.3}$ | $60.0_{\pm 0.2}$ | $58.6_{\pm 0.5}$ | $58.7_{\pm 0.3}$ |
| + MI (**TMCL**) | $60.4_{\pm 0.4}$ | $\mathbf{60.3}_{\pm 0.1}$ | $\mathbf{60.0}_{\pm 0.5}$ | $\mathbf{59.4}_{\pm 0.4}$ |

**The strength of MI controls the stability-plasticity trade-off.** We first investigate how different methods perform before and after observing task samples. Therefore, we introduce backwards and forward transfer metrics that measure the difference in task performance pre- and post-session compared to a model that is trained solely on that particular task using VI (definitions provided in the supplementary material). We observe that MI strongly improves both backwards and forward transfer compared to pure VI, while SupCon only provides mediocre improvements in forward transfer and even degrades backwards transfer (Table 4). Introducing a model state invariance term significantly enhances backward transfer, demonstrating positive knowledge transfer from previously learned tasks. Remarkably, the combination of MI and SI on average achieves performance within 3 percentage points of the task-specific baseline model, demonstrating significant forward transfer. To further investigate the effect of MI, we vary its strength $\lambda_{MI} \in [0, 1]$ in the combination VI + $\lambda_{MI}$MI. In Figure 3, for each session $s$, we observe the accuracy of test samples associated with classes from $\mathcal{C}^{(s)}$ during the course of training. We observe that increasing $\lambda_{MI}$ enhances both forward transfer (left of the gray region) and backward transfer (right of the gray region), albeit at the expense of current task performance (gray region).

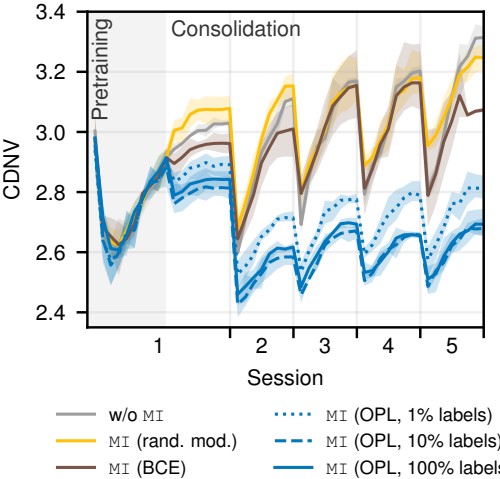

Figure 4: **CDNV during consolidation.** Measured on the CIFAR-100 test split on runs observing all labels (averaged over four seeds, shading indicates min. and max.)

Table 5: **Ablations.**

(a) **Ablating the role of orthogonalizing modulations.** Linear readout accuracy on continual CIFAR-100 and ImageNet-100 observing all labels (average, $\pm$ denotes the standard deviation).

| Method | CIFAR-100 | ImageNet-100 |
|---|---|---|
| VI + MI (**TMCL**) | $60.7_{\pm 0.4}$ | $63.5_{\pm 0.4}$ |
| untrained mods. | $60.1_{\pm 0.5}$ | $60.6_{\pm 1.5}$ |
| random mods. | $59.7_{\pm 0.5}$ | $60.9_{\pm 0.7}$ |
| OPL $\rightarrow$ BCE | $59.5_{\pm 0.5}$ | $63.8_{\pm 0.6}$ |
| w/o MI | $59.3_{\pm 0.2}$ | $59.7_{\pm 0.3}$ |

(b) **Ablating the role of architectural choices.** Linear readout accuracy on continual CIFAR-100 (averaged over four seeds, $\pm$ denotes the standard deviation).

| Method | CIFAR-100 |
|---|---|
| VI + MI (**TMCL**) | $60.7_{\pm 0.4}$ |
| w/o pred. | $55.1_{\pm 0.1}$ |
| w/o stop-grad | $60.3_{\pm 0.3}$ |
| w/o pred. & stop-grad | $60.3_{\pm 0.1}$ |

**Orthogonalized modulations improve class-separation.** We hypothesized in Section 3 that BCE is suboptimal, since negatives are collapsed towards a single vector. As we replace the orthogonal projection loss with binary cross-entropy, we observe degraded performance on CIFAR-100 (Table 5b). On ImageNet-100, performance is similar, which we attribute to the fact that the BCE training does not fully converge to an antiparallel configuration: On CIFAR-100, the average minimum BCE loss over all classes is $0.005_{\pm 0.039}$, while on ImageNet, it is $0.053_{\pm 0.035}$. Interestingly, while untrained modulations (i.e. frozen after normal initialization) as well as random modulations (i.e. redrawn each time) degrade performance further, they still provide a moderate improvement over pure VI. This suggests that predicting representations based on a slightly perturbed model state already provides a regulatory effect. We further measure the class-distance normalized variance (CDNV, appendix E) of representations [97], which is the ratio between intra-class variance and class-mean distances. A lower CDNV thus indicates a combination of higher intra-class collapse and/or higher distances between classes. We observe that modulation invariance improves class-separation as the CDNV significantly decreases once training enters the consolidation stage, but only in combination with orthogonalizing modulations (Figure 4). Moreover, the effect is still clearly visible while using 1% labels.

Table 6: **Results on ResNet-18.** Reporting linear evaluation performances on continual CIFAR-100 (averaged over four seeds, $\pm$ denotes the standard deviation).

| Method | Labeled frac. | |
|---|---|---|
| | 100% | 10% |
| VI | $53.4_{\pm 0.1}$ | |
|   + MI (**TMCL**) | $58.1_{\pm 0.1}$ | $56.7_{\pm 1.1}$ |
|   + SupCon | $54.5_{\pm 0.2}$ | $54.6_{\pm 0.5}$ |
|   + SI (**PNR**) | $59.9_{\pm 0.3}$ | |
|   + SI (**CaSSLe**)[a] | $60.1_{\pm 0.4}$ | |
|   + SI (**PNR**)[a] | $60.3_{\pm 0.4}$ | |
|   + ER[b] [61] | $54.6$ | |
|   + DER[b] [48] | $55.3$ | |
|   + LUMP[b] [96] | $57.8$ | |
|   + Less-Forget[b] [51] | $56.4$ | |
|   + POD[b] [56] | $55.9$ | |
| CLS-ER [78] | $56.7_{\pm 0.2}$ | $49.8_{\pm 0.2}$ |

[a] adapted from Cha et al. [72]
[b] adapted from Fini et al. [69]

**Architectural ablations.** The predictor network appears to be essential for MI, but only if a stop-gradient is imposed upon the modulated branches (Table 5b). Removing the stop-gradient while keeping the modulations frozen results in a moderate performance degradation, regardless of the presence of a predictor. Still, the stop-gradient is useful from both a computational perspective – reducing memory footprint and gradient computations – and from a biological perspective, as the

predictor with stop-gradient could be implemented by a hypothesized cortical predictive coding pathway [40].

**ResNet architecture.** As we replace ConViT with ResNet-18 (Table 6), we observe that SI significantly improves performance over standard VI, underlining the susceptibility of ResNets to forgetting compared to ViT-based architectures [98]. Introducing MI also significantly improves performance over VI as well as other label-integrating methods (SupCon, CLS-ER), but slightly underperforms SI. We hypothesize that the ability to directly modulate spatial relationships in ViTs lends more expressivity to the modulations in these models. Furthermore, the reliance of ResNet architectures on BatchNorms hinders stable training of different modulations, as it requires freezing the batch statistics. This is observable especially in the high standard deviation in accuracy of TMCL given 10% labels. As the underlying data statistics change with each session, the BatchNorms will adapt accordingly, and we hypothesize that this interferes with previously learned modulations. This is not the case for LayerNorm — as used by transformer architectures – which normalizes across features and does not depend on stored normalization statistics.

## 5   Limitations

Our work focuses on developing an understanding of algorithms that leverage the cortical circuitry, which is characterized by top-down and bottom-up information streams [99], to learn in naturalistic scenarios. Our analysis demonstrates this on a standard CIFAR-100 class-incremental setup, but omits data- and domain-incrementality. TMCL relies on static modulations, resulting in memory requirements that scale with the number of classes. For 100 classes, this amounts to around 4.1M parameters, while CaSSLe and PNR store a copy of the network (10.7M, Table A1b). Still, alternatives such as pruning simple classes or, better yet, a network that generates such modulations, should be explored. Finally, we observe that representational quality on the trained dataset (CIFAR-100) only moderately improves as more labels are presented, underperforming SupCon in the fully labeled regime. Still, given the improved transfer performance of TMCL, we hypothesize that this is indeed biologically plausible as the unmodulated network is not primed to solve CIFAR-100 specifically, but rather driven towards generalizable representations. Top-down priming could be implemented via a separate set of task-specific modulations, as has been explored in previous work [34–36].

## 6   Discussion

In this work, we have proposed a novel, brain-inspired algorithm for continual learning. It has been proposed that modulations provide a powerful framework for task-incremental learning, as they allow a general feature detector to learn new tasks by adapting modulations only [34, 36]. We extend this framework to class-incremental learning, and show that modulations can be consolidated into a shared representation space, sharpening percepts from data classes observed asynchronously. This consolidation co-opts the general machinery for view invariance learning, which in the brain is thought to be available anyway for predictive coding [15, 16]. Furthermore, there appear to be measurable parallels between our algorithm and cortical learning, as large-scale brain imaging indicates that representations of new stimuli orthogonalize from all others throughout learning, and unsupervised pretraining affects task learning [100]. Elucidating the drivers of this orthogonalization will provide further insight into how the brain leverages its biophysical machinery to achieve continual learning.

As we have shown that training for modulation invariance imparts task-specific information on the unmodulated network, one tantalizing possibility is that the effective learning objectives for networks, or parts of networks, could be tuned by incorporating sets of modulations that solve specific tasks. In a mixed language-vision approach [27], this could afford a fine-grained control over the eventual representation learning beyond what is currently possible with e.g. multi-task datasets [101]. In neurobiology, this represents a new view on intra-cortical and thalamocortical interactions. As cortical regions are targeted by modulations originating from distinct sets of brain areas with specific roles [102], the precise configuration of modulatory afferents could provide an understanding of the effective area-specific learning objectives. In turn, as these connectivity patterns are driven by genetically determined cues, this theory may afford insight into the distinct roles of genetics and plasticity in developing functional brains [103].

## Acknowledgements

This work was funded by the Federal Ministry of Education and Research (BMBF) under grant no. 01IS22094E WEST-AI as well as by Helmholtz Association's project-oriented funding programme (PoF 2, Topic 3). Furthermore, the authors gratefully acknowledge computing time on the supercomputers JURECA [104] at Forschungszentrum Jülich under grant no. jinm60. The authors also gratefully acknowledge the Gauss Centre for Supercomputing e.V. (www.gauss-centre.eu) for funding this project by providing computing time through the John von Neumann Institute for Computing (NIC) on the GCS Supercomputer JUWELS [105] at Jülich Supercomputing Centre (JSC). The authors also extend their gratitude to Sven Krauße for providing thoughtful comments on the manuscript.

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

# A Implementation Details

Table A1: **Methodological differences.**

(a) **Learning protocol.** Number of epochs are given per session.

|  | **Ours** | [69, 72] |
|---|---|---|
| $n_{\text{views}}$ | 4 | 2 |
| Pretraining epochs | 250 (CIFAR-100), 200 (ImageNet-100) | - |
| Consolidation epochs | 200 | 500 |
| Orthogonalization epochs | 100 | - |

(b) **Architectures.** Architecture numbers assuming $32 \times 32$ image inputs and 100 classes.

|  | **ConViT (DyTox)** [65, 95] | **ResNet-18** [41] |
|---|---|---|
| Parameters | 10.7M | 11.2M |
| + modulations | 4.1M | - |
| FLOPS | 1.4G | 1.4G |

For all CIFAR-100 experiments, we use the same class split as in CaSSLe, i.e. the same across all seeds. For the ImageNet-100 as well as the 10 session experiments (Section F), we use different class splits per seed.

For ConViT experiments, we use the AdamW optimizer [106] with a batch size of 256 and feedforward weight decay of $0.0001$ for all experiments, using a per-session cosine learning rate decay with 10 warmup epochs. ConViT experiments are trained with a feedforward learning rate of $0.001$ and a modulation learning rate of $0.01$. The ResNet experiments are trained with a feedforward learning rate of $1.0$ and a modulation learning rate of $0.3$ using the LARS optimizer ($\eta = 0.02$). The Barlow Twins losses are scaled down by a factor of $0.1$ for ConViT experiments, and by $0.025$ for ResNet experiments. We pick the redundancy-reduction weighting factor $\lambda_{\text{BT}} = 0.005$ for all experiments.

The ConViT backbone has 5 'local' self-attention blocks, replacing the self-attention layers with gated-positional self-attention layers, followed by a 'global' self-attention block with standard self-attention. We use 12 attention heads and a model dimension of $384$. All images are resized to input size 32 and we use a patch size of 4. The ResNet backbone conforms to the original ResNet-18, except that the first convolution layer has a kernel size of 3 and padding 2, the first MaxPool is removed, and we remove the final MLP layers.

The code for our experiments is available at `https://github.com/tran-khoa/tmcl`.

**Modulations.** The gain and bias modulations are applied to the query, key, value and output projections of all multi-head attention modules, and to both layers of the feedforward MLPs that follow the attention modules. Additionally, we modulate the positional prior of the ConViT-specific gated-positional self-attention (GPSA) layers, i.e. we modulate the operation $\mathbf{v}_{\text{pos}}^{h\,\text{T}}\mathbf{r}_{ij}$ (cf. Equation 7 from d'Ascoli et al. [95]). Gain and bias modulations are initialized from a Gaussian distribution, respectively from $\mathcal{N}(1, 0.02)$ and $\mathcal{N}(0, 0.02)$. We impose weight decay on the modulations with decreasing strength for deeper layers,

$$\texttt{weight-decay}(l) := 0.4 - 0.36(\cos(\pi \cdot l/L)) + 1)/2 \tag{7}$$

for modulations of the $l$-th ViT layer (out of $L = 6$ layers). Only random horizontal flips are used as augmentations for the modulations.

**Orthogonalization** We virtually constrain the number of batches to the actual number of samples divided by the batch size. Let $\mathcal{C}_t$ be the set of classes available at session $t$. For each batch, we sample uniformly (with replacement) $c \sim \mathcal{C}_t$. Let $X_c$ be the set of training samples of class $c$ and $X_{\neg c}$ be all other samples available at session $t$. Then, with probability 0.5, each class is sampled uniformly from $X_c$, or class $X_{\neg c}$ otherwise.

$$P(x_t = x) = 0.5 \cdot P(x \sim \text{Uniform}(X_c)) + 0.5 \cdot P(x \sim \text{Uniform}(X_{\neg c})). \tag{8}$$

**Consolidation** The projector $h$ is a three-layer MLP (dimensions 2028, 2048, 2048) with ReLU activation and BatchNorm in the hidden layers. The predictor $p$ for MI, CaSSLe and PNR is a two-layer MLP (dimensions 2048, 2048) with ReLU activation and BatchNorm in the input layer. All projectors and predictors are reset at the end of each session.

For SupCon, we use a two-layer MLP (dimensions 2048, 128) with ReLU activation and BatchNorm in the input layer, and we use a temperature of $0.1$ in the softmax.

As augmentations, we use

```
RandomResizedCrop(
    size=(32),
    scale=(0.08, 1.0),
    ratio=(3.0 / 4.0, 4.0 / 3.0),
    resample=Resample.BICUBIC,
),
ColorJitter(
    brightness=0.4,
    contrast=0.4,
    saturation=0.2,
    hue=0.1,
    p=0.8,
),
RandomGrayscale(p=0.2),
RandomHorizontalFlip(p=0.5),
RandomSolarize(p=0.2, thresholds=0.0, additions=0.0),
```

although Solarize is only applied for even views ($v \bmod 2 = 0$).

**Linear probing and k-nearest neighbors**    For linear probing, we follow standard methodology for self-supervised learning with vision transformers [107], i.e. we train a linear classifier on top of the `[CLS]` tokens from the four last layers on all training samples, regardless of the labels available during continual representation learning. For ResNets, we use the output of the last layer. We train for 100 epochs using stochastic gradient descent with momentum (batch size 1024, base learning rate 0.1 with cosine decay). We do not use any augmentations except for random horizontal flips. For k-nearest neighbors (kNN), we obtain the representations of the last layer of the backbone instead. No augmentations are used. The prediction is obtained by considering $k = 20$ nearest neighbors, weighted by distance with temperature $t = 0.07$.

**Compute**    We run our experiments on the JUWELS-Booster [115] and JURECA [116] clusters at Forschungszentrum Jülich. For both systems, we use a single NVIDIA A100 GPU per experiment. We observe empirically that SupCon based methods have the highest GPU memory consumption (22 GB), while modulation invariance methods use 17 GB. View and state invariance require 14 GB of GPU memory. Augmentations are run on GPU and the datasets do not require image decoding, therefore CPU and RAM requirements are negligible. All runs take up to 10 hours to finish on the five session scenario.

# B    Forward and Backward Transfer

## B.1    Metrics

Let task $i$ be the classification problem on the classes from session $i$. We then define $A_{t,i}$ as the evaluation accuracy of task $i$ at the end of training session $t$. We evaluate this accuracy in the task-agnostic setting, i.e. the classifier (linear or kNN) is unaware that the input data is limited to the task under consideration. In our FT and BT metrics, $\hat{A}_i$ is the task-agnostic kNN evaluation accuracy of task $i$ on a model trained from scratch using Barlow Twins on data from task $i$. Then, we define:

**Backward Transfer**

$$\text{BT} = \frac{1}{T-1} \sum_{i=1}^{T-1} \frac{1}{T-i} \sum_{t=i+1}^{T} (A_{t,i} - \hat{A}_i) \tag{9}$$

**Forward Transfer**

$$\text{FT} = \frac{1}{T-1} \sum_{i=2}^{T} \frac{1}{i-1} \sum_{t=1}^{i-1} (A_{t,i} - \hat{A}_i) \tag{10}$$

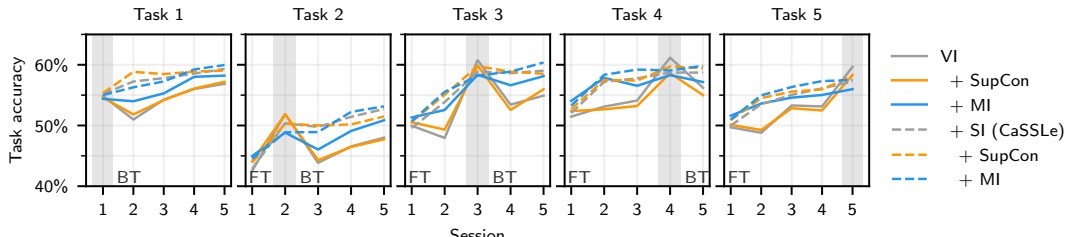

Figure A1: **Forward and backward transfer of different methods.** Accuracies on class-incremental CIFAR-100 (5 sessions) given either 1% of labels or completely unsupervised (averaged over four seeds).

## B.2 Forward and Backward Transfer of Different Methods

Previously, we investigated the stability-plasticity trade-off as we modify the strength of the modulation invariance term (Figure 3). We further demonstrate in the sparsely labeled learning scenario (1% labels), that modulation invariance also provides improved forward and backward transfer compared to SupCon, while SupCon surprisingly shows lower plasticity than VI (Figure A1). The introduction of state invariance shows further improvements, as the combination of SI and MI on top of VI yields the highest forward and backward transfer on most tasks except for the first task.

## C Python-style Pseudocode for TMCL

```
# f(xs, t): forward pass of backbone with inputs xs and modulations t
# C_1, ..., C_t: list of classes of session 1, ..., t
def orthogonalization(xs, ys):  # sparse labeled samples at session t
  for step in range(orthog_steps):
    c = random.sample(C_t, k=1)
    positives = random.sample(xs[ys == c], k=batch_size // 2)
    negatives = random.sample(xs[ys != c], k=batch_size // 2)

    pos_f, neg_f = f(positives, t=c), f(negatives, t=c)
    loss = opl_loss(pos_f, neg_f)
    loss.backward()
    update(f.modulations[c])

def consolidation(xs, is_pretrain=False): # unlabeled samples at session t
    for step in range(cons_steps):
        batch = random.sample(xs, k=batch_size)
        views = [aug(batch) for _ in range(num_views)]

        # view invariance
        # h_vi: view-inv. projector
        vi_projs = [h_vi(f(v, t=None)) for v in views]
        vi_loss = contrastive_loss(*vi_projs)  # Multi-view Barlow Twins

        if is_pretrain:
            vi_loss.backward()
            update(f.feedforward_weights)
            continue

        # if enabled: model state invariance
        # f_past: frozen backbone from prev. session
        # h_vi_past: frozen view-inv. projector from prev. session
        # p_si: state-inv. predictor
        with torch.no_grad():
            si_projs_past = [h_vi_past(f_past(v, t=None))]
```

```
        si_preds_curr = p_si(vi_projs)
        si_loss = mean(
            distill_loss(curr, past)  # CaSSLe/PNR loss function
            for curr, past in zip(si_preds_curr, i_projs_past)
        )

        # modulation invariance
        # h_mi: mod-inv. projector
        # p_mi: mod-inv. predictor
        mi_tasks = [[None] * batch_size] # unmodulated first view
        mi_tasks += [random.sample(C_1 + ... + C_t, k=batch_size) for _ in views[1:]]
        mi_projs = [h_mi(f(v, t=t)) for v, t in zip(views, mi_tasks)]
        mi_pred = p_mi(mi_projs[0])
        mi_loss = contrastive_loss(mi_pred, *mi_projs[1:])  # Multi-view Barlow Twins

        loss = vi_loss + si_loss + mi_loss
        loss.backward()
        update(f.feedforward_weights)

def train(sessions):
    for session_idx, (xs_unlabeled, xs_labeled, ys_labeled) in sessions:
        if session_idx == 0:
            consolidation(xs_unlabeled, is_pretrain=True)
        orthogonalization(xs_labeled, ys_labeled)
        consolidation(xs_unlabeled)
```

## D  Implicit Orthogonalization via Modulation Invariance

This section seeks to walk through the intuition behind the implicit orthogonalization via modulation invariance. To do so, we assume a non-incremental fashion with four classes $A, B, C, D$. For conceptual clarity, we focus the explanation here on the class centers of these respective classes.

In the orthogonalization phase of TMCL, we train modulations $m_A$ to achieve $A|_{m_A} \perp \{B|_{m_A}, C|_{m_A}, D|_{m_A}\}$, $m_B$ to achieve $B|_{m_B} \perp \{A|_{m_B}, C|_{m_B}, D|_{m_B}\}$, $m_C$ to achieve $C|_{m_C} \perp \{A|_{m_C}, B|_{m_C}, D|_{m_C}\}$, and $m_D$ to achieve $D|_{m_D} \perp \{A|_{m_D}, B|_{m_D}, C|_{m_D}\}$. Here, $A|_{m_A}$ denotes the representational vector of the class center of class A under modulation $m_A$, and similar for all others.

In the consolidation phase, our goal is to arrive at a representation space where $A|_\emptyset \perp B|_\emptyset \wedge A|_\emptyset \perp C|_\emptyset \wedge A|_\emptyset \perp D|_\emptyset \wedge B|_\emptyset \perp C|_\emptyset \wedge B|_\emptyset \perp D|_\emptyset \wedge C|_\emptyset \perp D|_\emptyset$ (where $A|_\emptyset, B|_\emptyset, C|_\emptyset$ and $D|_\emptyset$ denote the class centers of $A, B, C, D$ in the unmodulated network). It can be seen that this will be the case, if we simultaneously achieve the orthogonality relations of point 1 and $A|_\emptyset = A|_{m_A} = A|_{m_B} = A|_{m_C} = A|_{m_D}$ (where $A|_{m_{A,B,C,D}}$ denotes the class center of class $A$ respectively under modulations $m_A, m_B, m_C,$ and $m_D$) and similar for classes B, C, D. The contrastive objective maximizes similarity between a given data sample without modulation and under modulations $m_A, m_B, m_C,$ and $m_D$, and therefore implicitly drives the network to a state for which $A|_\emptyset = A|_{m_A} = A|_{m_B} = A|_{m_C} = A|_{m_D}$ and similar for $B, C, D$.

We note that achieving the full orthogonality relation $A|_\emptyset \perp B|_\emptyset \wedge A|_\emptyset \perp C|_\emptyset \wedge A|_\emptyset \perp D|_\emptyset \wedge B|_\emptyset \perp C|_\emptyset \wedge B|_\emptyset \perp D|_\emptyset \wedge C|_\emptyset \perp D|_\emptyset$ would likely require iterating steps 1 and 2. However, we did not seek such an iterating implementation, as we focused on the continual learning setting where we identified modulation learning (step 1) with a single phase of fast learning that occurs whenever a new class is observed, and step 2 with a slower consolidation phase. Under these conditions, the full orthogonality relation cannot be expected to be achieved rigorously, but as is demonstrated by our CDNV results, our TMCL algorithm still leads to a representation space where the clustering of the individual classes is improved.

# E Class-Distance Normalized Variance

The **class-distance normalized variance** (CDNV) is defined as

$$\text{CDNV} := \frac{1}{|\mathcal{C}|^2 - |\mathcal{C}|} \sum_{\substack{c,c' \in \mathcal{C} \\ c \neq c'}} \overbrace{\frac{\boxed{\text{Var}(\mathbf{Z}^{(c)}) + \text{Var}(\mathbf{Z}^{(c')})}}{\underbrace{2 \left\| \mu(\mathbf{Z}^{(c)}) - \mu(\mathbf{Z}^{(c')}) \right\|}_{\text{inter-class distance}}}}^{\text{intra-class collapse}}, \tag{11}$$

where $\mathbf{Z}^{(c)} = [\mathbf{z}_1^{(c)}, \ldots, \mathbf{z}_N^{(c)}] \in \mathbb{R}^{N \times D}$ is a batch of backbone representations (i.e. $\mathbf{z}_i^{(c)} = f(\mathbf{x}_i^{(c)} | \mathbf{W}, \emptyset)$) of all class-$c$ samples in the CIFAR-100 test split, and $\mathcal{C}$ is the set of CIFAR-100 classes (lower is better).

# F Split-CIFAR-100 with 10 sessions

We demonstrate that TMCL is also effective in the scenario of sparsely supervised continual learning with 10 sessions, i.e. 10 classes per session (Table A2).

Table A2: **Semi-supervised continual representation learning.** Final all-vs-all accuracy on class-incremental CIFAR-100 (10 sessions) given either 1% of labels or completely unsupervised, averaged over four seeds ($\pm$ denotes the standard deviation).

| Method | kNN | linear |
|---|---|---|
| SupCon + SI (PNR) | $34.7_{\pm 0.6}$ | $20.9_{\pm 1.4}$ |
| CE | $39.2_{\pm 0.8}$ | $28.2_{\pm 0.9}$ |
| VI | $55.5_{\pm 0.5}$ | $50.3_{\pm 0.4}$ |
| + MI (**TMCL**) | $57.7_{\pm 0.2}$ | $52.2_{\pm 0.3}$ |
| + SupCon | $55.1_{\pm 0.3}$ | $49.9_{\pm 0.4}$ |
| + SI (PNR) | $57.4_{\pm 0.5}$ | $54.0_{\pm 0.9}$ |
| + SupCon | $57.5_{\pm 0.2}$ | $54.0_{\pm 0.6}$ |
| + CE | $57.4_{\pm 0.3}$ | $54.2_{\pm 0.8}$ |
| + MI | $\mathbf{58.7}_{\pm 0.3}$ | $\mathbf{54.6}_{\pm 0.5}$ |

# G Datasets

Our analysis focuses on the CIFAR-100 dataset [117] and the ImageNet-100 dataset [111]. For the transfer learning experiments, we perform kNN evaluation on `Aircraft` [118], `CIFAR-10` [117], `CUBirds` [122], `DTD` [109], `EuroSAT` [113], `GTSRB` [114], `STL-10` [110], `SVHN` [119], and `VGGFlower` [120].

# H Further acknowledgements

We implement our methods based on PyTorch [108] with the Lightning framework [112]. For the augmentations on CIFAR-100, we used kornia [121]. Backbone implementations are adapted from the timm library [123]. Data loading and augmentations for the ImageNet-100 experiments are implemented via NVIDIA DALI (https://github.com/NVIDIA/DALI). For the figures, we relied on icons designed by OpenMoji (License: CC BY-SA 4.0).

## Supplementary References

[108] Jason Ansel, Edward Yang, Horace He, Natalia Gimelshein, Animesh Jain, Michael Voznesensky, Bin Bao, Peter Bell, David Berard, Evgeni Burovski, Geeta Chauhan, Anjali Chourdia, Will Constable, Alban Desmaison, Zachary DeVito, Elias Ellison, Will Feng, Jiong Gong, Michael Gschwind, Brian Hirsh, Sherlock Huang, Kshiteej Kalambarkar, Laurent Kirsch, Michael Lazos, Mario Lezcano, Yanbo Liang, Jason Liang, Yinghai Lu, CK Luk, Bert Maher, Yunjie Pan, Christian Puhrsch, Matthias Reso, Mark Saroufim, Marcos Yukio Siraichi, Helen Suk, Michael Suo, Phil Tillet, Eikan Wang, Xiaodong Wang, William Wen, Shunting Zhang, Xu Zhao, Keren Zhou, Richard Zou, Ajit Mathews, Gregory Chanan, Peng Wu, and Soumith Chintala. PyTorch 2: Faster Machine Learning Through Dynamic Python Bytecode Transformation and Graph Compilation. In *29th ACM International Conference on Architectural Support for Programming Languages and Operating Systems, Volume 2 (ASPLOS '24)*. ACM, April 2024.

[109] Mircea Cimpoi, Subhransu Maji, Iasonas Kokkinos, Sammy Mohamed, and Andrea Vedaldi. Describing textures in the wild. In *Proceedings of the IEEE conference on computer vision and pattern recognition*, pages 3606–3613, 2014.

[110] Adam Coates, Andrew Ng, and Honglak Lee. An analysis of single-layer networks in unsupervised feature learning. In *Proceedings of the fourteenth international conference on artificial intelligence and statistics*, pages 215–223. JMLR Workshop and Conference Proceedings, 2011.

[111] Jia Deng, Wei Dong, Richard Socher, Li-Jia Li, Kai Li, and Li Fei-Fei. Imagenet: A large-scale hierarchical image database. In *2009 IEEE conference on computer vision and pattern recognition*, pages 248–255. Ieee, 2009.

[112] William Falcon and The PyTorch Lightning team. PyTorch Lightning, March 2019.

[113] Patrick Helber, Benjamin Bischke, Andreas Dengel, and Damian Borth. Eurosat: A novel dataset and deep learning benchmark for land use and land cover classification. *IEEE Journal of Selected Topics in Applied Earth Observations and Remote Sensing*, 12(7):2217–2226, 2019.

[114] Sebastian Houben, Johannes Stallkamp, Jan Salmen, Marc Schlipsing, and Christian Igel. Detection of traffic signs in real-world images: The German Traffic Sign Detection Benchmark. In *International Joint Conference on Neural Networks*, number 1288, 2013.

[115] Stefan Kesselheim, Andreas Herten, Kai Krajsek, Jan Ebert, Jenia Jitsev, Mehdi Cherti, Michael Langguth, Bing Gong, Scarlet Stadtler, Amirpasha Mozaffari, et al. Juwels booster–a supercomputer for large-scale ai research. In *International Conference on High Performance Computing*, pages 453–468. Springer, 2021.

[116] Dorian Krause and Philipp Thörnig. Jureca: modular supercomputer at jülich supercomputing centre. *Journal of large-scale research facilities JLSRF*, 4:A132–A132, 2018.

[117] Alex Krizhevsky, Geoffrey Hinton, et al. Learning multiple layers of features from tiny images. 2009.

[118] Subhransu Maji, Esa Rahtu, Juho Kannala, Matthew Blaschko, and Andrea Vedaldi. Fine-grained visual classification of aircraft. *arXiv preprint arXiv:1306.5151*, 2013.

[119] Yuval Netzer, Tao Wang, Adam Coates, Alessandro Bissacco, Baolin Wu, Andrew Y Ng, et al. Reading digits in natural images with unsupervised feature learning. In *NIPS workshop on deep learning and unsupervised feature learning*, volume 2011, page 4. Granada, 2011.

[120] Maria-Elena Nilsback and Andrew Zisserman. Automated flower classification over a large number of classes. In *2008 Sixth Indian conference on computer vision, graphics & image processing*, pages 722–729. IEEE, 2008.

[121] E. Riba, D. Mishkin, D. Ponsa, E. Rublee, and G. Bradski. Kornia: an open source differentiable computer vision library for pytorch. In *Winter Conference on Applications of Computer Vision*, 2020.

[122] Catherine Wah, Steve Branson, Peter Welinder, Pietro Perona, and Serge Belongie. The caltech-ucsd birds-200-2011 dataset. 2011.

[123] Ross Wightman. PyTorch Image Models.

