# OpenReview forum: "Contrastive Consolidation of Top-Down Modulations Achieves Sparsely Supervised Continual Learning"
_NeurIPS.cc/2025/Conference — NeurIPS 2025 poster_

### Official Review · Reviewer_vrza · 2025-07-03

**Clarity:** 2
**Significance:** 3
**Originality:** 3
**Rating:** 5
**Confidence:** 4

**Summary:**

This paper tackles the problem of catastrophic forgetting, especially in cases where models need to learn from both unsupervised and supervised data. To solve this problem, this paper introduces task-modulated contrastive-learning (TMCL). TMCL has a feedforward backbone that used contrastive loss for unsupervised learning. TMCL also has a network that is trained to learn modulations to implement supervised learning of object identities (while the feedfoward backbone is frozen). To learn modulation-invariant representations, the feedforward network is subsequently trained to with a contrastive loss to bring together representations of differently modulated views of the same image and push those representations away from the unmodulated representation of another image. They test TMCL against state-of-the-art alternative models, and their model performs in the same ballpark (often better than the other models, but generally within a few percentage points).

**Questions:**

To make this easier to follow and respond to during the rebuttal, I’ve ordered my questions and suggestions to correspond with the points raised in the Weaknesses section above.

1. Model clarity

     a) I'd like to see the model explained fully and more clearly earlier in the paper. This would help readers have context around Figures 1 & 2 and even Related Work. I realize that it is rare to put a model description before Related Work, but page 5 is quite late to fully explain the model, and readers might be better equipped to put the Related Work in context if they already understand the model.
     b) I suggest revising Figure 2 to make it easier for readers to follow.
     c) What is the architecture of the modulated network?

2. Orthogonal representations of class identities

     Why does contrastive training of modulated views lead to an unmodulated representation space with orthogonal class identities? Can the authors walk this logic out in the paper more clearly?

3. Marginal performance gains

     Do the authors believe that their model is promising because of superior performance or because TMCL has novel biological inspiration and still performs comparably to state-of-the-art models? If the latter, I suggest explicitly making this argument in-text.

4. Confusing ablation results

     a) Why do untrained/random modulations perform nearly as well as trained modulations? The authors note that "This suggests that predicting representations based on a slightly perturbed model state already provides a regulatory effect." However, this does not sufficiently clarify why performance is so close to non-ablated VI+MI performance.
     b) The authors claim "Removing the stop-gradient while keeping the modulations frozen results in a moderate performance degradation," but that "moderate" degradation is only 0.7%. If the stop-gradient makes so little difference, does this call into question the necessity of a "hypothesized cortical predictive coding pathway"?
     c) Why is performance higher without pred. **and** stop grad than just without pred.?

One last question unrelated to the weaknesses: The paper claims that "information [arriving] at spatially segregated loci on sensory neurons [suggests] distinct roles in shaping the neuronal input-output relation" and distinct plasticity processes (page 3). Can the authors clarify why information arriving at different loci should imply different plasticity processes?

**Ethical Concerns:**

["NO or VERY MINOR ethics concerns only"]

**Final Justification:**

The authors clearly addressed the weaknesses from my review (see my last comment to the authors).

I also read through the other reviewers' comments to see if there were other critical issues, and my general summary isL
* Modulation scaling - To me, this is an interesting area for future research, which the authors appear to be working on based on their responses.
* CIFAR-100 - The authors added ImageNet experiments, which fully resolves this concern
* Comparison to other models - Personally, I don't see why having some biologically inspired models makes other biologically inspired models less interesting. This model strikes me as unique relative to models and baselines.
* Choices in the model that are not biologically inspired (batch setting) - I agree with the reviewer that this is a minor concern. This strikes me as an interesting future follow up idea. But we stifle model innovation if we expect papers initially introducing models to replace every ML technique with biological implementations. I think this contribution is sufficient without a streaming setting.

**Limitations:**

yes

**Quality:**

3

**Strengths And Weaknesses:**

## Strengths
- This paper tackles catastrophic forgetting, a long-standing and important issue in machine learning. Their approach to understanding catastrophic forgetting (noting that ecological learning is generally unsupervised with limited access to supervised labels) is also a refreshing approach.
- I appreciate the biological grounding of the model.  The authors draw inspiration from complementary learning systems theory, mapping fast learning to task-driven modulation and slow learning to unsupervised consolidation. This biological framing is conceptually interesting and helps motivate the model architecture. Ultimately, I don't think this *needs* to be the best performing model to be interesting. It is simply interesting that this different approach can still compete with other state-of-the-art models.
- I generally think that contrastive learning is a very promising approach. Contrastive learning has lost a little popularity as MAE approaches have gained popularity, but I'm still quite enthusiastic about the contrastive learning framework. This paper adapts contrastive learning in a novel way.
- TMCL performs comparably (and sometimes slightly better) than existing state-of-the-art models across several tasks.

## Weaknesses
1. One of my main concerns is clarity in terms of the model description. The high-level description of the model is difficult to follow, particularly early in the paper. Key details about how the system works only become clear around page 5 (lines 156–166), which might be too late to engage the reader effectively. Figure 2, while clearly intended to clarify the architecture, is difficult to parse. The relationship between the top and bottom panels is unclear, and labels like “Learning Pathway: modulations” do not meaningfully aid interpretation. I also do not have a confident grasp on what the architecture of the modulated network looks like.
2. The logic behind how contrastive training of modulated views leads to an unmodulated representation space with orthogonal class identities isn’t clearly justified. This seems to be an important theoretical claim, but it’s not adequately walked through. (The authors claim that this is true "intuitively," line 158).
3. Tables 2 & 3 suggest the model is performing similarly to others, with modest gains (~1–2%). This makes the paper feel like another iteration in the pattern of marginal improvement papers, rather than a conceptual leap forward.
4. The ablation results raise some questions and concerns for me. While it’s great to see ablations included, some of the results are unintuitive or potentially undermining. For example, untrained/random modulations perform nearly as well as trained ones. Also, performance is higher without pred. AND stop grad than just without pred. These results are not well explained and leave open questions about what components are actually contributing to the model’s success.

---

> ### Author Rebuttal · Authors · 2025-07-30
>
> We would like to thank the reviewer for their valuable feedback on our work. We are encouraged by the reviewer's appreciation of the biological grounding and the novel conceptual ideas of our manuscript. We improve the clarity of our model description and the intuition of the consolidation phase (W1, W2), and also provide new, more significant experimental results and ablations (W3, W4).
>
> ### 1. TMCL is primarily a machine learning analogue to our hypothesized cortical learning mechanism, but also provides significant improvements in learning *representations* continually. (W3/Q3)
> Our primary goal for this work was to achieve a conceptual algorithm that explains cortical learning. In particular, our algorithm aims to provide a computational framework that makes sense of the general cortical layout with top-down connections arriving distally on the dendritic trees of the neurons, representing modulations of the neural activity. Complementarily, we still aimed to demonstrate strong continual learning performance. We will make this hierarchy of objectives more clear in the introduction of the camera-ready version.
>
> ### 2. We will improve the high-level description of the model at the beginning of Section 3. (W1/Q1)
> We refer to the rebuttal to reviewer `XmrC` (Section 2), in which we provide the significantly improved description.
>
> ### 3. Performance improvements are significant with weaker ResNet baseline, and on the more complex ImageNet-100 dataset. (W3/Q3)
> On ImageNet-100, TMCL outperforms all other methods in the semi-supervised setting, improving over the `VI` baseline by +3.8pp (10% labels) and +2.3pp (1% labels). In the fully supervised setting, we observe TMCL improves over the `VI` baseline by +4.8pp on ImageNet-100 (`fD14`), and by +4.7pp on CIFAR-100 using ResNet-18 (Tab. 5 in the curr. manuscript).
> We refer to the rebuttals to `4K39` (Section 3) and `fD14` (Section 1, Tables 1 & 2).
>
> ### 4. We improve the explanation on how consolidating orthogonalized modulations introduces orthogonalization in the unmodulated representation space. (W1,2/Q1,2)
> We appreciate the feedback about the clarity of the exposition in our manuscript. Thanks to the reviewers' comments, we now realise that we should have been more clear in our description of the principles underlying our TMCL algorithm. We again refer to the rebuttal to reviewer `XmrC` (Section 2) for the improved explanation.
>
> Here, to walk out the logic in further detail, let us explain the idea in a non-incremental fashion with four classes $A, B, C, D$. For conceptual clarity, we focus the explanation here on the class centers of these respective classes.
>
>  1. In the orthogonalization phase of TMCL, we train modulations $m\_A$ to achieve $A\\vert\_{m\_A} \\bot \\{B\\vert\_{m\_A},C\\vert\_{m\_A},D\\vert\_{m\_A}\\}$, $m\_B$ to achieve $B\\vert\_{m\_B} \\bot \\{A\\vert\_{m\_B}, C\\vert\_{m\_B}, D\\vert\_{m\_B}\\}$, $m\_C$ to achieve $C\\vert\_{m\_C} \\bot \\{A\\vert\_{m\_C}, B\\vert\_{m\_C}, D\\vert\_{m\_C}\\}$, and $m\_D$ to achieve $D\\vert\_{m\_D} \\bot \\{A\\vert\_{m\_D}, B\\vert\_{m\_D}, C\\vert\_{m\_D}\\}$. Here, $A\\vert\_{m\_A}$ denotes representational vector of the class center of class A under modulation $m\_A$, and similar for all others.
>  2. In the consolidation phase, our goal is to arrive at a representation space where $A\\vert\_{\\emptyset} \\bot B\\vert\_{\\emptyset} \\wedge A\\vert\_{\\emptyset} \\bot C\\vert\_{\\emptyset} \\wedge A\\vert\_{\\emptyset} \\bot D\\vert\_{\\emptyset} \\wedge B\\vert\_{\\emptyset} \\bot C\\vert\_{\\emptyset} \\wedge B\\vert\_{\\emptyset} \\bot D\\vert\_{\\emptyset} \\wedge C\\vert\_{\\emptyset} \\bot D\\vert\_{\\emptyset}$ (where $A\\vert\_{\\emptyset}, B\\vert\_{\\emptyset}, C\\vert\_{\\emptyset}$ and $D\\vert\_{\\emptyset}$ denote the class centers of $A,B,C,D$ in the unmodulated network). It can be seen that this will be the case, if we simultaneously achieve the orthogonality relations of point 1 *and* $A\\vert\_{\\emptyset} = A\\vert\_{m\_A} = A\\vert\_{m\_B}  = A\\vert\_{m\_C} = A\\vert\_{m\_D}$ (where $A\\vert\_{m\_{A,B,C,D}}$ denotes the class center of class $A$ respectively under modulations $m\_A$, $m\_B$, $m\_C$, and $m\_D$) and similar for classes B, C, D. The contrastive objective maximizes similarity between a given data sample without modulation and under modulations  $m\_A$, $m\_B$, $m\_C$, and $m\_D$, and therefore implicitly drives the network to a state for which $A\\vert\_{\\emptyset} = A\\vert\_{m\_A} = A\\vert\_{m\_B}  = A\\vert\_{m\_C} = A\\vert\_{m\_D}$ and similar for $B,C,D$.
>
> We note that achieving the full orthogonality relation  $A\\vert\_{\\emptyset} \\bot B\\vert\_{\\emptyset} \\wedge A\\vert\_{\\emptyset} \\bot C\\vert\_{\\emptyset} \\wedge A\\vert\_{\\emptyset} \\bot D\\vert\_{\\emptyset} \\wedge B\\vert\_{\\emptyset} \\bot C\\vert\_{\\emptyset} \\wedge B\\vert\_{\\emptyset} \\bot D\\vert\_{\\emptyset} \\wedge C\\vert\_{\\emptyset} \\bot D\\vert\_{\\emptyset}$ rigorously would require iterating steps 1 and 2, and under certain conditions we expect a convergence proof can be constructed for this.  However, we did not seek such an iterating implementation, as we focused on the continual learning setting where we identified modulation learning (step 1) with a single phase of fast learning that occurs whenever a new class is observed, and step 2 with a slower consolidation phase. Under these conditions, the full orthogonality relation can not be expected to be achieved rigorously, but as is demonstrated by our CDNV results, our TMCL algorithm still leads to a representation space where the clustering of the individual classes is improved.
>
> ### 5. Orthogonalization ablations are significant on ImageNet-100. (W4/Q4)
> We believe that the combination of a stronger ConViT baseline as well as the simple CIFAR-100 dataset explain the small differences in ablation performances.
> Therefore, **we have now also performed the orthogonalization ablations on ImageNet-100, and observe much stronger differences under linear evaluation (Table 5)**.
> We will include these novel results in the camera-ready version to demonstrate the effectiveness of TMCL.
>
> ### 6. Interpretation of the ablation results. (W4/Q4)
> As for the specific points, regarding *(a)* we believe that maximising similarity over noise modulations enhances robustness of the representations, therefore leading to a higher performance than pure VI, in the same way as adding noise as an image augmentation also leads to higher performance (see e.g. [1]).
> Regarding *(b)*, we note that we consider the predictor + stop-grad as the more biologically plausible choice, as in a streaming setting the predictor could be interpreted as providing a representation originating from sensory input from the previous time point, allowing it to be compared to sensory input from the current time-point. Without stop-grad, synaptic plasticity would have to proceed based on two loss values at the same time, whereas having a neural implementation with only one loss value is already challenging (see e.g. [2]).
> Regarding *(c)*, at this point we only have these empirical observations, any statement we make would be speculation.
>
> ### 7. Distinct plasticity based on spatial location. (Q5)
> Broadly speaking, synaptic plasticity is governed by calcium influx, which happens when N-methyl-D-aspartate (NMDA) receptor channels in excitatory synapses are opened. These receptor channels need strong depolarization to open, which, at the soma, is mostly provided by action potentials, leading to classic spike-timing dependent plasticity. More distally in the dendritic tree, however, such depolarization is provided by local inputs, thus leading to very different plasticity dynamics. These effects are well documented in the neuroscience literature, and we suggest Kampa et al. [3] as a good starting point.
>
> ---
>
> **Table 5:** Ablations on ImageNet-100, averaged over 3 seeds
>
> | Method                            | kNN  | linear |
> | --------------------------------- | ---- | ------ |
> | VI                                | 50.5 | 59.7   |
> | **VI + MI (ours, TMCL)**                | 56.0 | 64.5   |
> | &nbsp;&nbsp;Untrained modulations | 50.7 | 60.6   |
> | &nbsp;&nbsp;Random modulations    | 50.5 | 60.9   |
> | &nbsp;&nbsp;OPL -> BCE            | 56.1 | 63.8   |
>
>
> #### References
> [1] Chen, et al., A simple framework for contrastive learning of visual representations. ICML 2020.
> [2] Sacramento et al., Dendritic cortical microcircuits approximate the backpropagation algorithm. NeurIPS 2018.
> [3] Kampa et al., Dendritic mechanisms controlling spike-timing-dependent synaptic plasticity. Trends in Neurosciences (10.1016/j.tins.2007.06.010)

---

> > ### Comment · Reviewer_vrza · 2025-08-06
> >
> > Thank you for your thorough responses to my review and the other reviewers. I found these responses incredibly helpful.
> >
> > Your rebuttal explanation cleared up my confusions from my first 2 concerns (model description and how the unmodulated representation space becomes orthogonal). In particular, your full explanation in the comment "We improve the explanation on how consolidating orthogonalized modulations introduces orthogonalization in the unmodulated representation space. (W1,2/Q1,2)" was especially clarifying. I recommend adding this detailed walk-through to SI. (I wish there were room in the main text, but 9 pages is likely too short.) I saw that the SI does have pseudocode, but I thought this was even clearer.
> >
> > I'm encouraged that TMCL is primarily a novel biologically inspired model, and even more impressed that it's performing so competitively on the new ImageNet experiments. And I'm encouraged that the ablation results are clearer for the ImageNet experiments as well.
> >
> > I also wanted to mention that I thought your conversation with reviewer fD14 about the modulation generator was very interesting. I'll be interested to read about this in future papers. And I look forward to reading the Kampa et al. paper you suggested.
> >
> > Overall, after reading your responses to my reviews and the other reviews, I am more enthusiastic about this paper, and I will adjust my score appropriately.

---

### Official Review · Reviewer_XmrC · 2025-07-03

**Clarity:** 2
**Significance:** 2
**Originality:** 2
**Rating:** 5
**Confidence:** 3

**Summary:**

Motivated by the continual learning ability of biological brains, this submissions considers the problem of learning good representations from mostly unlabelled data. The authors propose a representation learning method that uses a contrastive loss to learn modulations that orthogonalize representations. These modulations are then used to consolidate the data into the weights of the neural network. To evaluate the proposed method, the authors compare their method on different scenarios using CIFAR100, including on some downstream tasks.


Decision: My current assessment is a weak rejection with room for improvement. While the submission provides some insight and develops a generally applicable method for improving representation quality, it is held back by some clarity issues and experiments which are not entirely convincing.

**Questions:**

- The technical meaning of the term "modulation" and "top-down" is never defined. In equation (1) and (2) it seems to be defined as additional elementwise multiplicative and additive parameters, which play a specific role in representing task information. But I would prefer to see the difference from the usual weights more explicit. "BatchNorm [...] is equivalent to learning affine modulations" This adds to my confusion because the weights for batchnorm are treated as elementwise trainable weights, carrying no semantic meaning about task information.
- I see some discussion in reference to parameter-efficient fine tuning methods like refs 30 and 31. But what is the relationship between this method of adapting parameters to task and approaches that factorize matrices, such as low rank adapters?
- "modulations (‘How did we solve the task?’), and do not investigate methods with exemplar replay." -> it would still be interesting to compare results to replay methods for completeness.
- Are the modulations and weights trained in completely independent phases?
- If there are two phases, how is the data split between the phases? How can you train with the contrastive loss in the "task-learning" phase if it seems to require orthogonalization between classes, when the data is mostly unlabelled?
- I like the idea of orthogonalizing the modulations of each class, but I wonder if this representation may impede effective generalization in cases where it is not needed, such as in a single task-setting? (It is fine if this is the case, but it is not clear to me one way or another)
- Line 172: what is the rationale for using the average of the views here?
- Table 1: it is comparatively less clear what the set of modulations are, how they are sampled, and why there are an equal number of modulations (+1 for unmodulated) and augmentations (since both are indexed by $v$. Does every single instantiation of an augmentation (e.g., rotate by x degrees), correspond to a unique modulation m_x?
- Some theoretical justification for this approach would help clarify what the contribution of this method is beyond the presented motivation.

### Minor Comments
- Why is the term "session" used periodically when elsewhere the term "task" is used? They seem interchangeable, and should be standardized for clarity.
- line 177 (typo): appling -> applying
- Table 2 and 3 would be better served by emphasizing the proposed method, such as adding (Ours), to make the results stand out.
- $\mathcal{L}_{CL}$ is mentioned in the text but never defined, is it equivalent to $\mathcal{L}_{BT}$?

**Ethical Concerns:**

["NO or VERY MINOR ethics concerns only"]

**Final Justification:**

After the rebuttal, the authors have included additional results on a second dataset, which addresses one of my primary concerns. In their individual reply, the authors also clarified several points regarding the novelty of their approach (in consolidating different orthogonalizations). They also provided additional justification for the choices that they made during experiments (which are a departure from their original motivation) as well as described potential mechanisms to explain the effectiveness of learning modulations for orthogonalized representations.

Given the thorough reply, I will increase my score to a 5. However, the authors should update the submission with these additional details in a camera ready.

**Limitations:**

Yes

**Quality:**

3

**Strengths And Weaknesses:**

### Strengths
- The idea of learning transformations to orthogonalize the representation is compelling and seems effective in the experiments presented.
- Motivation: the authors make it clear that their motivated by understanding how learning algorithms can achieve the same continual learning ability as biological brains.
### Weaknesses
- While the experiments cover many different scenarios, the base dataset in all cases is CIFAR 100 can be limiting in its complexity and difficulty compared to ImageNet variants like tiny-ImageNet.
- It is not detailed exactly what this algorithm does. The loss and architecture are clear, but it is never put together in a way that makes it clear how the modulations are being learned and used.
- Minor but relevant given the motivation: Some choices in the submission add up to undercut the original biological motivation. For example, rather than a streaming setting as originally mentioned, the experiments are in the batch setting. Rather than alternating consolidation and task-learning, the model is given special phases in each task, one at the beginning for consolidation and then one for consolidation.

---

> ### Author Rebuttal · Authors · 2025-07-30
>
> We appreciate that the reviewer finds our biologically grounded idea of continually consolidating modulations into an orthogonalized representation space compelling, and thank the reviewer for their inquisitive remarks. We have reworked the introduction to Sec. 3 to improve clarity of the method early on in the paper, and address experimental concerns by providing new ImageNet-100 results.
>
> ## 1. New ImageNet-100 experiments show larger improvements. (W1)
> On ImageNet-100, TMCL outperforms all other methods in the semi-supervised setting by +3.8pp (10% labels) and +2.3pp (1% labels), cf. rebuttal to `fD14` (Sec. 1, Tab. 1 & 2).
>
> ## 2. We will improve our introduction to Section 3, explaining the integration of the loss and architecture. (W2)
> We will now start Section 3 (i.e. replace L109-114) with these paragraphs instead:
>
> > We follow the common idea from neuroscience [6-8] that cortical networks learn to interpret novel information by learning new top-down modulations, and propose that consolidation of these modulations is a crucial component of learning in biological brains. This motivates our task-modulated contrastive learning (TMCL) algorithm as the machine learning analogue of this consolidation, tackling continual representation learning. We consider the parameters of conventional machine learning models as *task-agnostic* feedforward weights $\\mathbf{W}$. On top of these weights, we introduce per-task affine transformation parameters as *task-specific* modulations $\\mathbf{m}\_t$, as an analogue of biological top-down modulations (Eq. 2 in the curr. manuscript). We denote the modulated network as $f(\\mathbf{x}|\\mathbf{W}, \\mathbf{m})$ with feedforward weights $\\mathbf{W}$ and modulations $\\mathbf{m}$, while $f(\\mathbf{x}|\\mathbf{W}, \\emptyset)$ represents the unmodulated network (i.e. where the modulations are identity operations).
>
> > In TMCL, the overall objective is to arrive at an unmodulated representation space where all classes $c \\in \\mathcal{C}$ in the dataset $\\mathcal{D}$ have compact representations clustered around mutually orthogonal class centers, i.e. $$
> \\gamma^c\\bot\\gamma^{c^{\\prime}},\\forall c,c^{\\prime}\\in\\mathcal{C}, c \neq c^{\\prime},(1)$$
> with $\\gamma^c = \\mathbb{E}\_{\\mathbf{x} \\in \\mathbf{X}^{(c)}}[f(\\mathbf{x} \\vert W, \\emptyset)]$, where $\\mathbf{X}^{(c)}$ is the set of samples from class $c$. Because we assume a continual learning setting, where we do not have all class samples at our disposal, we do not optimize for (1) directly. Rather, we achieve this by breaking the optimisation procedure down into two distinct learning objectives. The first objective is to **orthogonalize** any given class $c$ from the others in a *modulated* representation space, i.e. we learn a modulation $\\mathbf{m}^{c}$ so that the class center of class $c$ becomes orthogonal to all other classes in the modulated space:
> $$\\gamma^{c}\_{\\mathbf{m}^{c}} \\bot \\{ \\gamma^{c^{\\prime}}\_{\\mathbf{m}^{c}} : c^{\\prime} \\in \\mathcal{C} \\setminus \\{c\\} \\}, (2)$$
> where $\\gamma^c\_{\\mathbf{m}}$ denotes the representation of the class center under modulation $\\mathbf{m}$, i.e. $\\gamma^c\_{\\mathbf{m}} = \\mathbb{E}\_{\\mathbf{x} \\in \\mathbf{X}^{(c)}}(f(\\mathbf{x} \\vert W, \\mathbf{m}))$, via Eq. [3 in the curr. manuscript]. The second objective (**consolidation**) is entirely unsupervised and trains network weights to become *modulation-invariant*, so that $$
> \\gamma^c = \\gamma^c\_{m^{c^{\\prime}}} \\forall c^{\\prime} \\in \\mathcal{C}, (3)$$
> via [Eq. 5, Tab.1 in the curr. manuscript].
> It can be seen that a representation space for which *both* (2) and (3) hold, implies (1).
>
> > In our continual setting, which we adapt from Fini et al. [1], training is partitioned into $s\\in{1,\\ldots, S}$ sessions, and therefore (1) can only be achieved approximately. In each session $s$, we only observe unlabeled samples $x\\in\\mathcal{D}^{(s)}\\subset\\mathcal{D}$ belonging to the session-specific partition of classes $\\mathcal{C}^{(s)}\\subset \\mathcal{C}$. Additionally, few labeled samples $(x, y)\\in\\mathcal{D}^{(s)}\_\\text{sup}\\subset \\mathcal{D}^{(s)}$ are made available to (3). As a consequence, during each session (Fig. 2), we learn objective (2) restricted to $\\mathcal{D}^{(s)}\_\\text{sup}$ in a first phase, and then learn objective (3) using unlabeled samples from $\\mathcal{D}^{(s)}$. We explain the implementation of both phases in detail below.
>
> The combination of `VI + MI` will be explicitly marked as "TMCL" in the tables.
>
> ## 3. The batch setting is chosen for comparability to literature. (W3)
> In literature, continual self-supervised learning (SSL) is performed entirely in a batched setting. In fact, while we believe that SSL - as an implementation of predictive coding - is a biologically plausible mechanism, online self-supervised learning is an active area of research and out of scope for this work. Orthogonalization could be implemented in an online fashion, and remains to be explored in future work. We note finally that TMCL could be adapted straightforwardly to a setting where new classes appear asynchronously in the data stream, by replacing the negatives in the OPL with randomly draw samples from the unsupervised stream.
>
> ## 4. Questions
> ### On the terminology of "modulation" and "top-down". (Q1)
> With "top-down", we refer to the origin of the modulations in the biological setting, i.e. we expect modulations to originate from higher-order cortical areas, shaping neuronal behaviour based on high-level information (such as the current task). Indeed, we refer to the element-wise multiplicative and additive parameters as "modulations". These are separate from the task-agnostic weights, and in contrast to the weights, they are per-task.
>
> Regarding BatchNorm, we intended to express that **per-task** BatchNorms would be equivalent, resulting in per-task elementwise trainable shifts $\\gamma\_t$ and biases $\\beta\_t$ (for all tasks $t$).
>
> ### Modulations can be considered a parameter-efficient fine-tuning technique, but our main contribution is the *consolidation* of task-specific modulations. (Q2)
> We aim to clarify that **the novelty of our method is** not the idea of task-specific modulations for model adaptation (for which similar methods have been explored in literature), but **the consolidation of task-specific modulations**. In fact, TMCL would allow for the consolidation of "1-vs-all" orthogonalizing LoRA adapters, which is out of scope for our biologically grounded paradigm.
>
> ### We provide prior and new results with replay-based methods. (Q3)
> Our work builds upon continual representation learning literature, in particular CaSSLe [1], which investigate the replay-based methods ER [2] and DER [3] in a completely unsupervised setup (i.e. `VI + ER`, `VI + DER`), and find that these methods underperform `SI (CaSSLe)`.
> We obtain new results with CLS-ER [4] - a biologically-inspired method we compare to in our related work section - which underperforms our method `VI + MI` given all labels and deteriorates significantly when only provided 10% of the labels (cf. Tab. 4, rebuttal to `4K39`).
>
> ### Further questions
> > Are the modulations and weights trained in completely independent phases? **(Q4)**
>
> Yes, they are (as outlined in Figure 2 in the manuscript) made more clear in the new start of section 3).
>
> > If there are two phases, how is the data split between the phases? How can you train with the contrastive loss in the "task-learning" phase if it seems to require orthogonalization between classes, when the data is mostly unlabelled? **(Q5)**
>
> This is now explained in the new start of section 3.
>
> > [...] I wonder if this representation may impede effective generalization in cases where it is not needed, such as in a single task-setting? [...]
>
> We indeed expect modulation invariance (`MI`) to hurt generalization in a single task setting, or if the tasks do not cover the problem space sufficiently. Nevertheless, we empirically find that CIFAR-100 one-vs-all tasks cover the problem space sufficiently well to achieve strong generalization (Table 3 in the submitted manuscript).
>
> > Line 172: what is the rationale for using the average of the views here?
>
> While the idea of taking an average originates from a known multi-view version of Barlow Twins [5], the idea is compatible with a simple idea from vector symbolic architectures (VSA):
> For sample $\\mathbf{x}$ and class $c$, $f(\\mathbf{x}|\\mathbf{W}, \\mathbf{m}\_c)$ describes a feature vector - e.g. "x is dolphin", "x is not cat" and "x is not dog". The objective is to minimize the distance of the unmodulated representation $\\mathbf{v}=f(\\mathbf{x}|\\mathbf{W}, \\emptyset)$ towards all of these "feature vectors", for which the optimal solution is to pull $\\mathbf{v}$ towards the centroid of these feature vectors.
>
> > Table 1: it is comparatively less clear what the set of modulations are, how they are sampled, and why there are an equal number of modulations [...] and augmentations [...]
>
> For each view, we draw a separate augmentation (except for `SI`, following literature) as well as a separate modulation (`MI`).
>
> > Some theoretical justification for this approach would help clarify what the contribution of this method is beyond the presented motivation.
>
> We refer to the theoretical considerations we formulated in the rebuttal to `vrza`, as well as in the new start of section 3.
>
> ### Minor Comments
> - We will streamline the usage of session and task to make sure that "task" refers to the tasks solved by our one-vs-all modulations.
> - $\\mathcal{L}\_\\text{CL}=\\mathcal{L}\_\\text{MV-BT}$ for all methods except `SI`, for which we use $\\mathcal{L}\_\\text{CL}=\\mathcal{L}\_\\text{BT}$.
>
> ----
> ### References (cf. manuscript)
> [1] Fini et al. 2022
> [2] Robins. 1995
> [3] Buzzega et al. 2020
> [4] Arani et al. 2022
> [5] Ghosh et al. 2024
> [6] Iyer et al. 2022
> [7] Wybo et al. 2023
> [8] Gilbert et al. 2013

---

> > ### Comment · Reviewer_XmrC · 2025-08-07
> >
> > Thanks for the detailed response, this has helped clarify a few things. Two questions remain for me:
> >
> > >For each view, we draw a separate augmentation (except for SI, following literature) as well as a separate modulation (MI).
> >
> > I would think that a specific view is associated with a specific augmentation. Data augmentation typically involves resampling a new augmentations at each iteration. In your case, is there a finite set of augmentations considered? I ask because each augmentation is associated with a specific modulation, and it seems like there are only a finite set of modulations.
> >
> > > Relationship between orthogonalization and consolidation
> >
> > The reply to Reviewer vzra provides additional detail, but its still not clear the direction of influence for these two components of your approach. For example, is it orthogonalization that makes consolidation effective? Or is it that consolidation helps achieve orthogonalization in the unmodulated space? And if the second point is more pertinent, what advantage does this approach provide over directly orthogonalizing the unmodulated space?

---

> ### Author Response · Authors · 2025-08-07
>
> Dear Reviewer `XmrC`,
>
> We would like to thank you again for your time and your suggestions.
> - Based on your questions, we have **improved our introduction to section 3** to clarify our algorithm to the reader.
> - As you and other reviewers requested, we provide **new results on ImageNet-100**, showing **stronger, significant improvements** than previously observed on CIFAR-100.
>
> We hope that our answers and updates have resolved your main concerns, and are available for any final questions.

---

### Official Review · Reviewer_4K39 · 2025-07-03

**Clarity:** 2
**Significance:** 2
**Originality:** 2
**Rating:** 3
**Confidence:** 5

**Summary:**

The paper presents a biologically inspired ML method for continual learning. The paper combines contrastive learning and orthogonal representation learning to mitigate forgetting and learn from partially labelled data in a largely unsupervised data stream. Results on the CIFAR-100 dataset show improvement over other baseline CL methods.

**Questions:**

Please see the weaknesses section for my questions.

**Ethical Concerns:**

["NO or VERY MINOR ethics concerns only"]

**Final Justification:**

I have engaged in extensive discussions with the authors and looked at other reviewers' discussions. I noticed that all the reviewers are in agreement to accept the paper, and agree that the biologically inspired concepts presented by the paper are novel and interesting for continual learning. While I agree that the paper is interesting and these concepts are novel, it is still unclear why we need another paper that provides an interesting idea but fails to clarify how it might advance the continual learning field as a whole and provide a viable direction to deal with the fundamental problems in continual learning.

Due to these reasons, I am inclined to reject the paper, though I would not strongly oppose it being published.

**Limitations:**

Limitations have been discussed adequately. However, positive and negative impacts have not been discussed. This should be added in the paper as there can be a variety of impacts that can be discussed for the proposed method.

**Quality:**

2

**Strengths And Weaknesses:**

Strengths:
1) The paper covers the biological concepts quite well and provides a strong motivation for biologically inspired learning methods for continual learning.
2) The paper considers the problem of semi-supervised continual learning, where most of the data is unlabelled, which is closer to realistic settings.
3) A large number of experimental results are reported on CIFAR-100 that show improvement compared to a few other models.

Weaknesses:
1) While the model does combine some well-known concepts into a new method, how does it compare to other biologically inspired methods? For example, SI [b], FearNet [a], CBCL [c], [d, e], to name a few. There have been a lot of CL methods proposed that have been biologically inspired. How does the current method compare to those methods? And more importantly, what is the novelty of the proposed approach compared to prior works?
2) The proposed work has only been compared against a few baselines from the literature. How does this model compare against well-known CIL methods, such as replay-based, regularization-based, generative replay-based, etc., methods?
3) From tables 2 and 3, I do not see significant changes in performance by adding different modules to VI. This might mean that many of these modules are not contributing much to the overall performance, and it could be because of using the ViT architecture compared to a CNN, such as ResNet-18. How does the model perform with ResNet-18? The paper states in L202-23 that ViT was chosen because of its ability to mitigate forgetting. However, ViT is not the main contribution of this work, so it would be interesting to see the contribution of the backbone compared to the proposed biologically inspired modules.

[a] FearNet: Brain-Inspired Model for Incremental Learning, ICLR 2020.
[b] Zenke, F., Poole, B., & Ganguli, S. (2017, July). Continual learning through synaptic intelligence. In International conference on machine learning (pp. 3987-3995). PMLR.
[c] Ayub, A., & Wagner, A. R. (2023). Cbcl-pr: A cognitively inspired model for class-incremental learning in robotics. IEEE Transactions on Cognitive and Developmental Systems, 15(4), 2004-2013.
[d] Madireddy, S., Yanguas-Gil, A., & Balaprakash, P. (2023, November). Improving performance in continual learning tasks using bio-inspired architectures. In Conference on Lifelong Learning Agents (pp. 992-1008). PMLR.
[e] Wang, L., Zhang, X., Li, Q., Zhang, M., Su, H., Zhu, J., & Zhong, Y. (2023). Incorporating neuro-inspired adaptability for continual learning in artificial intelligence. Nature Machine Intelligence, 5(12), 1356-1368.

---

> ### Author Rebuttal · Authors · 2025-07-30
>
> We thank the reviewer for their extensive feedback. We believe the reviewer's concerns regarding novelty and comparison to previous methods stem from misunderstandings. We will improve our manuscript to alleviate the confusion, and we invite the reviewer to read the updated introduction to Section 3 (`XmrC`, Sec. 2). We fully address performance concerns with new experiments on ImageNet-100 and previous results on ResNet-18.
>
> ### 1. TMCL is not just a combination of prior "modules". (W1)
> As to our knowledge, the concept of learning to be invariant to modulations - carrying high-level "top-down" information - is novel in both neuroscience and machine learning. We acknowledge that at first glance, Table 1 in the manuscript suggests TMCL to be a combination of prior methods. However, combinations with `SI` solely suggest that `MI` is not orthogonal to prior approaches. **The main contribution is the introduction of consolidation of previously learned modulations via modulation invariance (`MI`), in combination with view invariance (`VI`), i.e. `TMCL = VI + MI`**. We will alleviate this confusion by simplifying Table 1 in the manuscript and moving complex combinations to a separate table.
>
> ### 2. TMCL is primarily a machine learning analogue to our hypothesized cortical learning mechanism, but also provides significant improvements in learning *representations* continually. (W1, W2, W3)
> From previous work on cortical learning, in particular the role of NMDA-driven top-down modulations impinging on distal dendritic branches and therefore shaping feedforward computations, we derive **a biologically inspired machine learning analogue**.
> TMCL aims to tackle continual *representation* learning in the spirit of CaSSLe [1] and PNR [2]. While these works - and our work - evaluate on class-incremental benchmarks, **these are not *class-incremental learning* algorithms**, since the readout (either linear or kNN) is obtained on all available data and is hence considered part of the evaluation. In Section 6, we discuss the omission of previous class-incremental approaches, which have already been demonstrated to be inferior to CaSSLe [1].
>
> ### 3. Performance improvements are significant with weaker ResNet baseline, and on the more complex ImageNet-100 dataset. (W3)
> We would like to stress that **we do not primarily aim to outperform state-of-the-art methods**, but to show that our biologically grounded method is at least competitive to the state-of-the-art (as Reviewer `vrza` identified).  However, as the reviewer correctly points out, the changes in performance are less significant since we opted to focus on the stronger ConViT baseline.
>
> **Still, we point out that we do in fact provide results on ResNet-18 in the manuscript.**
> We observe a significant improvement of TMCL (`VI + MI`) over `VI` on ResNet18 (**+4.7pp in linear evaluation**, cf. Table 5 in the submitted manuscript). However, the improvements of `VI + SI (PNR)` are stronger with ResNet18 (+6.5pp), which we attributed to the susceptibility of ResNets to forgetting, as the reviewer points out. We want to highlight that these results suggest that TMCL works on both ViT and ResNet architectures.
> **Significant improvements over `VI` are also observed on ConViT on ImageNet-100 (Rebuttal to `fD14`, Section 1, Tables 1 & 2),** as TMCL outperforms all other methods in the semi-supervised settings, improving over the `VI` baseline by +3.8pp (10% labels) and +2.3pp (1% labels).
>
> ### 4. Suggested methods [a,c-e] are not comparable because they aren't primarily representation learners. We instead provide results to a comparable biologically inspired method. (W1)
> We furthermore thank the reviewer for pointing out the references to these continual learning approaches that draw inspiration from biology **(weakness 1)**. It was not fully clear to us whether the reviewer is most interested in comparing the performances achieved by these various models, or rather in discussing their biological merits. As for performance, we considered SI [b] to be similar to elastic weight consolidation (EWC), which was evaluated in the CaSSLe paper [1] and underperforms both theirs and our `VI + MI` results (Table 4). FearNet [a], CBCL [c], and NNA [d] use a feature extractor pretrained on ImageNet. These approaches can thus not be considered representation learning algorithms, as they focus on learning a readout (or readout network) on top of a powerful pretrained model. Finally, [e] appears to only report task-incremental performance numbers. Instead, we compare to CLS-ER as a strong, biologically-inspired continual learning method (Table 4). TMCL outperforms CLS-ER given all labels, while CLS-ER significantly deteriorates in scarce supervision regime (10% labels).
>
> ### 5. Suggested biologically-inspired methods [a-e] are complementary to TMCL. (W1)
> As for biological relevance, we note that the biological mechanics these papers aim to leverage in a continual learning setup are complementary to ours. SI [b] focuses on preserving synaptic weights important for prior tasks, while FearNet [a] proposes hippocampal and prefrontal modules for generative replay. Finally, CBCL [c] uses the attractive cognitive idea of storing pseudo-class centroids based on a clustering algorithm. Our paper describes a novel mechanism based on the spatial organisation of the cortical microcircuit. The remarkable performance of biological systems in processing continual data streams is likely a combination of several of such mechanisms. Therefore, a systematic study investigating the merits of combining several of these biologically plausible continual learning mechanisms would certainly be interesting, but is outside the scope of the present work. It should be noted, however, that when one aims to extend generative replay mechanisms, or centroid based mechanism, to full sensory representation learning (i.e. beyond the learning of a shallow readout), one naturally arrives at a view where these mechanisms modulate the neurons in the sensory network. Therefore, using these approaches in some way as the input to a modulation generation network is a promising extension of our current work.
>
> ### 6. CaSSLe / PNR baselines cited in the manuscript are state-of-the-art for continual representation learning. (W2)
> Fini et al. performed studies on continual representation learning on a set of standard class-incremental approaches, namely "prior-focused regularization (EWC), data-focused regularization (POD, Less-Forget), and rehearsal-based replay (ER, DER) methods", and found these to be inferior to CaSSLe. For more details, refer to [1]. Therefore, similarly to [2], we assume CaSSLe [1] / PNR [2] to be state-of-the-art, but will provide results from [1] via Table 4 in the final version of our paper. Again, we want to highlight that these methods are complementary to TMCL.
>
> ### 7. Impact Statement (regarding "Limitations")
> With TMCL, we develop a novel, biologically-inspired algorithm for continual representation learning. Our work elucidates potential learning mechanisms in the cortex, contributing to the neuroscientific understanding of predictive coding as a canonical learning rule and the biological machinery behind the orthogonalization of stimuli. At the same time, TMCL allows for semi-supervised representation learning, e.g. in robotics and edge devices in a more realistic continual learning setting in which unlabelled samples are vast and labels are sparse. We do not find any direct negative impacts beyond the ones posed by fundamental machine learning and neuroscience research.
>
> ---
> **Table 4:** Comparison to CLS-ER and prior class-incremental approaches on ResNet-18 and CIFAR-100, averaged over 4 seeds. We adapt further results from [1].
>
> | Method (R18)                                   | linear   |
> | ---------------------------------------------- | -------- |
> | CLS-ER                                         | 56.7     |
> | CLS-ER (10%)                                   | 48.8     |
> | VI                                             | 53.4     |
> | &nbsp;&nbsp; + **MI** (ours, TMCL)             | 58.1     |
> | &nbsp;&nbsp; + **MI** (ours, 10% labels, TMCL) | 56.7     |
> | &nbsp;&nbsp; + ER [1]                          | 54.6     |
> | &nbsp;&nbsp; + DER [1]                         | 55.3     |
> | &nbsp;&nbsp; + LUMP [1]                        | 57.8     |
> | &nbsp;&nbsp; + Less-Forget [1]                 | 56.4     |
> | &nbsp;&nbsp; + POD [1]                         | 55.9     |
> | &nbsp;&nbsp; + SI (CaSSLe) [1]               | **60.4** |
>
> #### References
> [1] Fini et al., Self-supervised models are continual learners. CVPR 2022.
> [2] Cha et al., Regularizing with pseudo-negatives for continual self-supervised learning., ICML 2024.

---

> > ### Comment · Reviewer_4K39 · 2025-08-04
> >
> > Thank you for responding to my comments and questions.
> >
> > While some parts have been clarified, mainly that some of the prior cognitively inspired methods use pre-trained networks, while the proposed method does not, many concerns remain. Mainly, the authors state that the purpose of the paper is not to beat SOTA performance. While beating SOTA is not a requirement, there must be other significant contributions to support why the method might be important for the CL community and how it could advance this field. Since there has been a large amount of research already on biologically inspired methods, it is still unclear what this paper would contribute to the CL literature. Also, concepts such as orthogonalization in the representation space have been exhaustively explored in a variety of CL methods before, although not necessarily under the biologically inspired domain. Based on this, and other reviewers' comments, I maintain my score.

---

> > > ### Author Response · Authors · 2025-08-05
> > >
> > > We thank the reviewer for the continued engagement. We understand that now, your main concern is about the contributions of TMCL to continual learning in general.
> > >
> > > ### TMCL *is* state-of-the-art in continual semi-supervised representation learning with sparse labels.
> > > - Despite SOTA not being our primary target, our method **outperforms all other semi-supervised methods (SupCon, CE) in sparsely labeled continual representation learning setups** on CIFAR-100 (Tab. 2), transfer learning based on CIFAR-100 (Tab. 3) and **with significant margins on ImageNet-100** as well as on ResNet-18 (Rebuttal to this reviewer, Sec. 3).
> > > - Thanks to reviewer `fd14`, we now additionally show that TMCL is the only semi-supervised method that is **robust to significant label noise** (`fd14`, Sec. 3).
> > >
> > >
> > > ### Our main contribution is *not* the goal of orthogonality, but a biologically grounded way to implicitly achieve it.
> > >
> > > - Specifically, we propose that in biological brains, predictive coding over top-down modulations carrying high-level information shapes synaptic plasticity.
> > > Our continual machine learning analogue - TMCL - achieves all-vs-all orthogonalization **implicitly** via invariance learning over task-specific modulations that before, have been trained explicitly on one-vs-all class orthogonalization, and we show this empirically (Fig. 4).
> > > We believe that this new paradigm is a significant contribution in itself.
> > >
> > > - To our knowledge, prior work enforces orthogonality explicitly by orthogonalizing weight subspaces for different tasks [1] or orthogonalizing task gradients [2-4], with the goal of mitigating network overlap [5]. In contrast, we aim to introduce transfer learning between the different modulations, enabling forward and backward transfer as empirically demonstrated in the manuscript (Fig. 3, Tab. 4). This is conceptually different to the traditional understanding of orthogonalization.
> > >
> > >
> > > #### References
> > > [1] Chaudry et al., Continual Learning in Low-rank Orthogonal Subspaces. NeurIPS 2020
> > > [2] Zeng et al., Continual learning of context-dependent processing in neural networks. Nature Machine Intelligence, 2019.
> > > [3] Farajtabar et al., Orthogonal gradient descent for continual learning. AISTATS 2020
> > > [4] Saha et al., Gradient projection memory for continua learning. ICLR 2021
> > > [5] van de Ven et al., Continual Learning and Catastrophic Forgetting. arXiv:2403.05175

---

> > > > ### Comment · Reviewer_4K39 · 2025-08-06
> > > >
> > > > Thank you for responding to my comments and continuing the discussion.
> > > >
> > > > **SOTA Performance** The authors earlier stated that SOTA performance was not the main purpose of this work. They later stated that the method does achieve SOTA performance on one of the continual learning settings, beating **two** other methods. I think it is a bit of an overstatement that the proposed method beats all other SOTA continual learning methods. So, coming back to the earlier point I stated, what is the contribution of this paper to the CL literature?
> > > >
> > > > While I appreciate the explanation of orthogonalization of the paper compared to prior works, how does this new definition help with continual learning, theoretically and empirically, compared to other use cases of orthogonalization in prior methods? Additionally, predictive coding has also been explored for continual learning. For example [a] (I can provide more references if you like).
> > > >
> > > > [a] Song, Y., Millidge, B., Salvatori, T., Lukasiewicz, T., Xu, Z., & Bogacz, R. (2024). Inferring neural activity before plasticity as a foundation for learning beyond backpropagation. Nature neuroscience, 27(2), 348-358.

---

> > > > > ### Author Response · Authors · 2025-08-07
> > > > >
> > > > > We again thank the reviewer for the continued engagement.
> > > > >
> > > > > 1. **Predictive coding as a learning objective, not a solution to the credit assignment problem**. Unlike [a], we do not explore predictive coding as a replacement to backpropagation. Our main contribution is not the idea of introducing predictive coding to continual learning, as we outline in the next point.
> > > > > 2. We ourselves argue that predictive coding has been explored in the context of continual self-supervised learning in the form of *view invariance* (and CaSSLe, PNR further introduce *state invariance* to mitigate forgetting). TMCL extends these principles to incorporate high-level information in the form of sparse labels. **Our core contribution is the idea that predictive coding of *top-down one-vs-rest modulations* effectively orthogonalizes the representation space, which allows continual learning of sparsely labelled classes.** To our knowledge, this idea is novel.
> > > > > 3. While prior "orthogonalization-based" methods assume explicit task labels to find orthogonal task subspaces, our orthogonalization proceeds implicitly through the one-vs-rest modulations, and does not require external task labels.
> > > > > 4. **SOTA to underline concrete and significant performance benefits in continual machine learning, not the primary goal.** As to our knowledge, PNR is state-of-the-art in continual unsupervised representation learning (as of ICML 2024). As outlined in Sec. 6 of our rebuttal, both CaSSLe and PNR outperform prior continual learning methods that have been adapted towards representation learning. Therefore, for sparse labels, we consider comparing to "supervised learning + CaSSLe/PNR" as an appropriate benchmark for semi-supervised learning.

---

> > > > > > ### Comment · Reviewer_4K39 · 2025-08-09
> > > > > >
> > > > > > Thank you for your response to my comments.
> > > > > >
> > > > > > The main answer given to my question about the contribution of the paper by the authors is, "Our core contribution is the idea that predictive coding of top-down one-vs-rest modulations effectively orthogonalizes the representation space, which allows continual learning of sparsely labelled classes. To our knowledge, this idea is novel." I agree that this particular way of using predictive coding seems novel, but again, what is the contribution of the paper? What is this particular way of looking at continual learning contributing to the literature and advancing the research area as a whole? How does it contribute to the fundamental questions in continual learning: stability, plasticity, and computationally efficient transfer of past knowledge? Given these concerns, I maintain my score.

---

### Official Review · Reviewer_fD14 · 2025-07-03

**Clarity:** 2
**Significance:** 3
**Originality:** 3
**Rating:** 5
**Confidence:** 2

**Summary:**

This paper proposes a biologically inspired approach for semi-supervised continual learning, called Task-Modulated Contrastive Learning (TMCL). Inspired by the segregated cortical architecture of top-down and feedforward pathways, the method uses modulations (learned affine transformations) to integrate sparse supervisory signals into a contrastive learning framework. The key insight is that top-down modulations learned from few labeled examples can be consolidated into a modulation-invariant representation space through predictive coding-like contrastive losses. This allows learning new concepts with minimal forgetting, even in class-incremental learning without task identity. The method is extensively evaluated on CIFAR-100 under varying supervision regimes and transfer learning tasks.

The paper is generally well-written.

**Questions:**

1- can authors add a discussion on how to scale TMCL to larger datasets or class hierarchies?

2- can authors provide a visualization of representation space dynamics across sessions (e.g., UMAP plots) to support the modulation-invariance hypothesis?

3- can authors discuss robustness to label noise or incorrect modulations?

**Ethical Concerns:**

["NO or VERY MINOR ethics concerns only"]

**Final Justification:**

The authors have thoroughly addressed my concerns and I appreciate their detailed responses to other reviewers' concerns.

- The authors' response to the comparison of TMCL with other benchmark models clarifies the contribution of the work. Like reviewer vrza, I believe that having this biologically inspired model doesn't make other biologically inspired models less interesting. Moreover, adding this biological details expands the impact of this work to other fields such as systems neuroscience.

- Biological brains effortlessly solve continual learning problem. TMCL has comparative performance to other methods using brain inspired algorithm. This is done through orthogonalization via invariance learning inspired by cortical predictive learning.

- Biological brains face uncertainty even when they are provided with labeled feedback. The authors have added new analysis assessing the robustness of TMCL to label noise which shows comparative advantages over previous methods such as SupCon.

**Limitations:**

yes

**Quality:**

3

**Strengths And Weaknesses:**

Strengths:The orthogonal projection loss (OPL) to train class-specific modulations, and the consolidation of those modulations using modulation-invariant contrastive learning (MI), is a contribution.TMCL performs competitively or better than existing state-of-the-art methods in low-label or no-label continual learning settings.

Weaknesses: -Since a unique modulation is stored per class, scaling to tasks like ImageNet-1K may pose challenges in both memory and management. This is briefly mentioned as a limitation, but a forward-looking strategy (e.g., shared modulation generators) would be helpful.

-All experiments are performed on CIFAR-100. This dataset is limited in visual diversity and resolution. Results on higher-resolution benchmarks like miniImageNet or ImageNet-Subset would improve generalizability.

---

> ### Author Rebuttal · Authors · 2025-07-30
>
> We appreciate that the reviewer highlights the novelty of orthogonalizing modulations and the consolidation thereof via modulation invariance. As suggested by most reviewers, we now provide results on ImageNet-100, showing stronger improvements of TMCL. We will further strengthen our manuscript thanks to the reviewer's discussion on scaling classes and robustness to label noise.
>
> ### 1. New ImageNet-100 experiments show strong improvements over baselines. (W2)
> We have now extended our empirical study to ImageNet-100 (5 sessions). Our experiments (cf. Tables 1, 2) show that our method (+ MI) outperforms all other methods on 10% and 1% labels, with a significant improvement of +3.8pp and +2.3pp given 10% and 1% of ImageNet-100 labels respectively on linear evaluation.
>
> ### 2. Per-class modulations are cheaper than sample replay, and replacing them with a modulation generator is a promising avenue for future work. (W1, Q1)
> In principle, the number of classes that can be accommodated in TMCL is unlimited. Of course, in our current setup, there is a practical limit in terms of memory. Analyzing the memory consumption (Table A1b) shows that storing modulations for 100 classes requires storing approximately 4.1M parameters, compared to 10.7M for the storing the base ConViT model (without modulations). This means that modulations for 250 classes take up as much memory as the base model, marking the break-even point with SI methods (CaSSLe, PNR), which store a copy of this base model.  Furthermore, this equates to around 16MB per 100 classes, while e.g. storing 500 ImageNet images (on average) already takes 50MB.
>
> However, an avenue of particular interest is the generation of modulations with a modulation generating network. In such a setup, memory size is fixed, and the number of classes that can be accommodated is determined by the expressivity of the modulation generator. In future work, we aim to explore such a setup, as this is anyway closer to how we hypothesize that modulations of sensory networks are being generated in the brain.
>
> An important consequence of having such a modulation generator, is that TMCL can be extended to incorporate class hierarchies. Modulations for narrow classes (for instance, types of birds) could be consolidated into a modulation for a more general class ("bird") through TMCL, which in turn could be consolidated into a yet more general class ("animal"). Recognition of sensory percepts could then proceed along this hierarchy (i.e. "type of object? - animal" --> "type of animal? - bird", "type of bird? - sparrow", where each of these questions represents a different modulation applied to the sensory system). Such a process, we believe, is close to how biological perception of unfamiliar objects works.
>
> ### 3. TMCL is very robust to label noise, while other supervised methods deteriorate. (Q3)
> The suggestion of the reviewer piqued our interest, and we ran an experiment testing the robustness of TMCL w.r.t. label noise. The state-of-the-art supervised continual representation learning method (`CE`) proposed by Marczak et al. [1] underperforms unsupervised Barlow Twins (`VI`) as label noise increases past 40%. Similarly, they show that at least 30% of the supervised data is necessary in order to compete with `VI`, while TMCL (via `MI`) does not deteriorate in regimes with significantly fewer labels.
> In these new experiments, we replace k% of labels with random ones (Table 3). We observe that `MI` is robust to almost completely random labels, and is the only loss that does not deteriorate performance below the unsupervised `VI` baseline. We note that this is consistent with our ablations where we use random or untrained modulations, since for high label noise levels, the modulations would carry no semantic meaning and just represent an additional noise source to which the self-supervised objective learns to become invariant. We note furthermore that the supervised objective never *directly* influences the unmodulated representation space, opposed to e.g. SupCon, where with high label noise the supervised learning has a direct detrimental effect on the representations. We will add these results with label noise to the appendix of the camera-ready version.
>
> ### 4. Class-Distance Normalized Variance (CDNV) quantifies representation space dynamics, suggesting orthogonalization via modulation invariance. (Q2)
> The reviewer requests a visualization of the representation space dynamics to support the hypothesis that modulation invariance consolidates the class orthogonalizations introduced by the modulations. Unfortunately, we are unable to attach these visualizations in this year's rebuttal modality. However, our analysis of the evolution of the CDNV demonstrates that class clustering improves during the consolidation phase with `MI`, but not with pure `VI` or any other methods.
>
> ----
> **Table 1:** Class-incremental ImageNet-100, 5 sessions (linear eval., averaged over 3 seeds)
> | **Method**                               | **100% labels** | **10% labels** | **1% labels** | **unsup.** |
> | ---------------------------------------- | --------------- | -------------- | ------------- | ---------- |
> | SupCon                                   | 64.0            | 48.7           | 33.6          | -          |
> | &nbsp;&nbsp; + SI (PNR)                  | 64.1            | 48.5           | 33.5          | -          |
> | CE                                       | 66.0            | 50.1           | 35.2          | -          |
> | VI                                       | -               | -              | -             | **59.7**   |
> | &nbsp;&nbsp; + SupCon                    | 66.6            | 61.4           | 60.2          | -          |
> | &nbsp;&nbsp; + CE                        | 64.5            | 61.4           | 60.3          | -          |
> | &nbsp;&nbsp; + **MI** (ours, TMCL)       | 64.5            | **63.5**       | **62.0**      | -          |
> | &nbsp;&nbsp;+ SI (PNR)                   | -               | -              | -             | 59.6       |
> | &nbsp;&nbsp;&nbsp;&nbsp; + SupCon        | **67.0**        | 60.2           | 43.9          | -          |
> | &nbsp;&nbsp;&nbsp;&nbsp; + CE            | 64.5            | 60.3           | 59.4          | -          |
> | &nbsp;&nbsp;&nbsp;&nbsp; + **MI** (ours) | 63.8            | 62.7           | 61.7            | -          |
>
> **Table 2:** Class-incremental ImageNet-100, 5 sessions (kNN eval., averaged over 3 seeds)
> | **Method**                               | **100% labels** | **10% labels** | **1% labels** | **unsup.** |
> | ---------------------------------------- | --------------- | -------------- | ------------- | ---------- |
> | SupCon                                   | 57.5            | 40.2           | 25.6          | -          |
> | &nbsp;&nbsp; + SI (PNR)                  | 57.6            | 40.7           | 25.7          | -          |
> | CE                                       | **62.8**        | 44.0           | 26.1          | -          |
> | VI                                       | -               | -              | -             | 50.5       |
> | &nbsp;&nbsp; + SupCon                    | 57.7            | 52.4           | 50.9          | -          |
> | &nbsp;&nbsp; + CE                        | 56.2            | 52.0           | 51.0          | -          |
> | &nbsp;&nbsp; + **MI** (ours, TMCL)       | 56.0            | 55.1           | 52.1          | -          |
> | &nbsp;&nbsp; + SI (PNR)                  | -               | -              | -             | **51.7**   |
> | &nbsp;&nbsp;&nbsp;&nbsp; + SupCon        | 59.3            | 52.4           | 51.9          | -          |
> | &nbsp;&nbsp;&nbsp;&nbsp; + CE            | 56.4            | 52.5           | 51.6          | -          |
> | &nbsp;&nbsp;&nbsp;&nbsp; + **MI** (ours) | 56.3            | **55.3**       | **52.7**      | -          |
>
> **Table 3:** Label noise (linear evaluation, averaged over 4 seeds)
>
> | Method             | 30% noise | 50% noise | 90% noise | 99% noise |
> | ------------------ | --------- | --------- | --------- | --------- |
> | CE                 | 57.2      | 54.4      | 8.4       | 7.9       |
> | SupCon             | 58.2      | 56.8      | 48.7      | 47.0      |
> | *VI (unsup.)*      | *59.3*    | *59.3*    | *59.3*    | *59.3*    |
> | VI + SupCon        | **60.9**  | 60.0      | 58.3      | 58.3      |
> | VI + CE            | 59.3      | 59.0      | 58.3      | 58.3      |
> | **VI + MI (ours, TMCL)** | 60.6      | **60.3**  | **60.3**  | **59.4**  |
>
>
> ### References
> [1] Marczak et al., Revisiting supervision for continual representation learning. ECCV 2024.

---

> ### Comment · Reviewer_fD14 · 2025-08-07
>
> I thank the authors for their thorough response to my comments. I still think there are is value in this work because it uses brain inspired algorithms to improve AI systems. Even though there might not be dramatic improvement in the performance of this approach compared to previous approaches, yet it teaches us there multiple ways to solve the same problem. I maintain my score.

---

### Note · Authors · 2025-08-13

We would like to take this opportunity to thank the reviewers for the discussion and the constructive feedback. To facilitate discussions, we summarize our contributions and main improvements.

### Summary of contributions
- We introduce a novel paradigm of cortical learning, extending predictive coding principles to integrate modulations that convey information from higher-order cortical areas [L33-61].
    - These are contributions to the neuroscientific understanding of (1) predictive coding as a canonical learning rule and (2) the mechanism behind the empirically observed orthogonalization of stimuli in biological brains. [L280-290]
- We spotlight the effectiveness of TMCL in a sparsely labeled class-incremental representation learning setting: by replaying and consolidating one-vs-all modulations for past classes, TMCL implicitly orthogonalizes representations of novel classes against these past classes. Therefore, TMCL directly addresses knowledge transfer instead of merely trying to avoid forgetting as in prior continual learning literature [L156-166, Sec. 2 of rebuttal to `XmrC`].
- TMCL outperforms prior state-of-the-art self-supervised methods (`VI + PNR`), comparable semi-supervised methods (e.g. `VI + PNR + SupCon`), as well as prior supervised methods (`CE`, `SupCon`) in continual representation learning settings with sparse supervision (10%/1% labels)[Tables 2, 3, 5; Sec. 1 of rebuttal to `fD14`].
- We suggest TMCL as a canonical learning paradigm beyond continual learning, as we suggest higher-order cortical areas (or machine learning equivalents, e.g. networks of different modalities) provide top-down modulations that steer canonical predictive coding-based plasticity [L291-301].

### Summary of improvements
In response to the reviewers' concerns, we have significantly improved our manuscript:
1. We significantly reworked the introduction to section 3 to **clarify TMCL and the implicit orthogonalization argument** (`XmrC`, Sec. 2)
2. We provide **new results on ImageNet-100, showing stronger, significant improvements than observed previously on CIFAR-100**, as well as clearer ablation results (`fD14`, Sec. 1), and
3. We **additionally show that TMCL is robust to label noise in comparison to other supervised loss terms**, i.e. standard cross-entropy and `SupCon` (`fD14`, Sec. 3).

We thank the reviewers and the area chair for their time and consideration.

---

### Decision · Program_Chairs · 2025-09-17

**Decision:**

Accept (poster)

**Comment:**

The paper introduces Task-Modulated Contrastive Learning (TMCL), a biologically inspired approach to semi-supervised continual learning that integrates sparse supervisory signals into a contrastive learning framework. By leveraging modulations learned from limited labeled examples, TMCL aims to learn modulation-invariant representations, mitigating catastrophic forgetting while learning new concepts in a task-agnostic way. TMCL is evaluated extensively on CIFAR-100 (and ImageNet-100 in rebuttal) and shows improvements over baseline methods in many scenarios.

There was significant variance in initial reviewer opinions of this submission. In particular, Reviewer 4K39 questioned the novelty and significance of the work to the broader continual learning community. Through a sequence of vigorous back-and-forth exchanges between the authors and Reviewer 4K39, the reviewer relaxed their overall objection to the work, and the authors convincingly argued that the approach is interesting both due to its biologically-inspired nature and the state-of-the-art results on the considered datasets. The novelty of the approach seems clear, based on the other reviewer comments regarding the implicit orthogonalization via invariance learning. As for the outstanding questions from Reviewer 4K39 regarding the significance of the contribution to the continual learning community, NeurIPS is a form for broader discussions on machine learning from diverse perspectives. This contribution is interesting it its combination of ideas into a novel, biologically grounded model for semi-supervised Continual Learning. It thus adds something to the discussion on continual learning, potentially bringing fresh ideas from the broader community to bear on its fundamental problems. The recommendation is thus to Accept.